# NashPG: A Policy Gradient Method with Iteratively Refined Regularization for Finding Nash Equilibria

## Abstract

Finding Nash equilibria in two-player zero-sum imperfect-information games remains a central challenge in multi-agent reinforcement learning. Recent multi-round regularization methods offer a promising direction, yet existing approaches either require full enumeration of the game tree or rely on non-policy-gradient inner solvers that underperform in practice, leaving a scalable policy-gradient-based solution open. In this paper, we propose a novel multi-round regularization procedure and show that it guarantees strictly monotonic reduction in Bregman divergence to Nash equilibria and eventual convergence to one in two-player zero-sum extensive-form games. Guided by this framework, we develop a practical algorithm, *Nash Policy Gradient* (NASHPG), which places the regularization directly in the policy optimization objective and is implemented using standard policy gradient methods. Empirically, NASHPG achieves comparable or lower exploitability than prior model-free methods on classic benchmark games and scales to large domains such as *Battleship* and *No-Limit Texas Hold'em*, where it attains comparable or higher average payoff in head-to-head play.

## 1 Introduction

The success of deep reinforcement learning has sparked significant interest in multi-agent reinforcement learning (MARL) (Albrecht et al., 2024), where algorithms must handle environments with multiple interacting agents. Within MARL, two-player zero-sum (2p0s) imperfect-information games (IIGs) are one of the most studied settings because they capture key challenges such as partial observability and competition, while remaining more tractable than general-sum multi-player games. In these games, Nash equilibria are the central solution concept where agents are unexploitable even against optimal adversaries. Our goal is to develop general solvers for 2p0s IIGs, specifically MARL algorithms that are model-free, converge to a Nash equilibrium, and scale to large domains.

Recently, a new class of regularization-based methods (Liu et al., 2023) has been proposed. These approaches add regularization terms to the learning objective (Hofbauer & Hopkins, 2005), which make the resulting objective strongly convex–concave, yielding efficient update rules with last-iteration convergence to a regularized equilibrium (McKelvey & Palfrey, 1995). However, a regularized equilibrium is not necessarily a Nash equilibrium, so the converged policy can remain exploitable. To recover Nash equilibria, several recent works have explored *multi-round* schemes that keep the regularization strength fixed while periodically updating the regularization terms (Perolat et al., 2021; 2022; Abe et al., 2024; Lu et al., 2025). This direction is appealing because it offers a route to exact Nash equilibria without abandoning the stabilizing effect of regularization.

However, the practicality of multi-round regularization methods remains underexplored. Most prior work focuses on normal-form games (NFGs) or small extensive-form games (EFGs), where exact feedback is available. To our knowledge, one of the few practical instantiations is Regularized Nash Dynamics (R-NaD) (Perolat et al., 2021). However, a recent study (Rudolph et al., 2025) shows that R-NaD underperforms policy gradient methods on imperfect-information benchmark games. We hypothesize that this surprising result is driven by differences in the maturity of the underlying policy's update method. R-NaD relies on

Neural Replicator Dynamics (NeuRD) (Hennes et al., 2020), which is less studied compared to policy gradient methods which have benefited from decades of work on stability and performance (we explicitly test this hypothesis in Section 5.4). This motivates the question: *Can we develop a policy-gradient-based algorithm that provably converges to a Nash equilibrium in 2p0s IIGs?*

We answer this question in two steps. We first introduce *Iterative MMD* (IMMD), a multi-round extension of Magnetic Mirror Descent (MMD) (Sokota et al., 2023), where each round uses the converged strategy profile from the previous round to define the regularization. MMD is a natural starting point because its mirror-descent-update for mixed strategies in EFGs naturally suggests a policy-gradient-update counterpart for stochastic policies in reinforcement learning context. Our main theoretical result shows that IMMD yields strictly monotonic reduction in Bregman divergence to Nash equilibria.

**Theorem 1.** *(Proof in Appendix A.5.1) For a 2p0s EFG, let $z^*$ be a Nash equilibrium with $z^* \in \mathrm{int}\,\mathrm{dom}\,\psi \cap \mathcal{Z}$. Starting from an initial strategy profile $z_0$, IMMD generates a sequence of strategy profiles $\{z_t\}$ satisfying*

$$B_\psi(z^*; z_t) > B_\psi(z^*; z_{t+1}), \tag{1}$$

*for all $t$ such that $z_t$ is not a Nash equilibrium, where $B_\psi$ denotes the Bregman divergence. Moreover, $\{z_t\}$ converges to a Nash equilibrium, i.e., $\lim_{t\to\infty} z_t$ is a Nash equilibrium.*

This result motivates a practical MARL design: retain the multi-round regularization structure of IMMD, and implement the policy update with policy gradient methods. We instantiate this idea in *Nash Policy Gradient* (NashPG), which places the regularization directly in the policy objective. This design allows NashPG to be implemented using standard *policy gradient* methods, such as PPO (Schulman et al., 2017). Empirically, we show that NashPG achieves comparable or lower exploitability than prior model-free methods on classic benchmark games including Leduc Poker, Dark Hex $3 \times 3$, and Phantom Tic-Tac-Toe. To demonstrate scalability, we further evaluate NashPG on large-scale domains such as *Battleship* and *Heads-Up No-Limit Texas Hold'em*, where it achieves higher average payoff in head-to-head play.

## 2 Related Work

Existing methods for solving 2p0s IIGs can be broadly grouped into four categories. The main axes relevant to this paper are whether a method is model-free, whether it relies on populations or average policies, and whether it targets exact Nash equilibria.

**Population-Based Methods.** Representative algorithms in this category include Neural Fictitious Self-Play (NFSP) (Heinrich & Silver, 2016) and Policy Space Response Oracles (PSRO) (Lanctot et al., 2017), along with more recent extensions built on the same paradigm (McAleer et al., 2024; 2021; Bighashdel et al., 2024). These methods iteratively compute best-responses against mixtures of prior policies using deep reinforcement learning (DRL) and can scale to complex domains such as StarCraft (Vinyals et al., 2019). However, they require repeated best-response computations, resulting in high computational overhead. Moreover, they only guarantee average-iterate convergence where policy extraction is non-trivial in DRL settings, often requiring supervised learning or the storage of historical agents to obtain a final policy (Lanctot et al., 2019). In contrast, NashPG does not require policy extraction or maintaining a growing population of policies, at any point it only keeps the current policy and the regularization reference policy.

**CFR-Inspired Methods.** Counterfactual Regret Minimization (CFR) (Zinkevich et al., 2007), most notable for its success in poker games, enabled superhuman performance in Heads-Up No-Limit Texas Hold'em (Brown & Sandholm, 2018). Deep CFR (Brown et al., 2019) integrates function approximation for larger state spaces. However, CFR relies on explicit game models for tree traversal, which limits its generalizability, and often relies on high-variance sampling to estimate counterfactual values. While some model-free variants do exist (Steinberger et al., 2020), they still suffer from large variance issues, and variance-reduction techniques (McAleer et al., 2023) can introduce exploration biases. Similar to population-based methods, CFR approaches typically guarantee only average-iterate convergence. Recent benchmarking shows that these model-free CFR methods underperform compared to MMD (Sokota et al., 2023) and R-NaD (Perolat et al., 2021) on IIG benchmarks (Rudolph et al., 2025); we therefore scope our experimental comparison to

the latter stronger baselines. In contrast, NASHPG is model-free and uses standard policy-gradient updates from sampled trajectories, avoiding high-variance counterfactual estimators and additional policy extraction.

**Regularization-Based Methods.** These methods (Liu et al., 2023) address last-iterate convergence by reformulating games as strongly convex–concave objectives, typically converging to entropy-regularized Nash equilibria such as Quantal Response Equilibria (McKelvey & Palfrey, 1995). Several recent works (Abe et al., 2023; Cen et al., 2021; Daskalakis & Panageas, 2019) establish strong convergence guarantees in this setting. Among them, Magnetic Mirror Descent (Sokota et al., 2023) is especially relevant to our theoretical development, as our analysis builds on the same variational inequality (VI) (Facchinei & Pang, 2003) perspective. However, regularized equilibria are not necessarily Nash equilibria, so the converged policy can remain exploitable. Our work builds on this foundation and uses a multi-round refinement procedure to target exact Nash equilibria.

**Multi-Round Regularization Methods.** Most relevant to our work are methods that periodically update a regularization reference policy. Within this line of work, a useful distinction is between *enumerative* methods, whose updates require full gradient information or evaluation over all decision points, and *sample-based* methods, which learn from sampled trajectories. The first group includes Adaptively Perturbed Mirror Descent (APMD) (Abe et al., 2024) and Divergence-Regularized Discounted Aggregation (DRDA) (Lu et al., 2025). Our proposed IMMD also belongs to this group, but differs in how regularization enters the update: IMMD adds regularization directly to the *payoff function*, whereas APMD perturbs the *gradient field*, and DRDA regularizes a discounted Follow the Regularized Leader (FTRL) (Shalev-Shwartz & Singer, 2006) style value dynamic. These methods are theoretically relevant, but their reliance on full enumeration makes them intractable in large domains, which motivates the need for practical sample-based methods.

Among sample-based methods, the closest prior work to NASHPG is R-NaD (Perolat et al., 2021), which combines multi-round regularization with Neural Replicator Dynamics (NeuRD) (Hennes et al., 2020). However, recent evidence shows that R-NaD underperforms policy gradient methods on IIGs benchmark (Rudolph et al., 2025), so this direction remains practically unresolved. NASHPG revisits the same multi-round regularization framework with a different inner optimization design, instead of incorporating regularization through reward transformation, it optimizes the regularized policy objective directly, allowing the method to be implemented using standard policy-gradient methods. Table 1 summarizes these distinctions.

| Algorithm | Inner update | Regularization enters | Update type | Guarantee |
|---|---|---|---|---|
| **IMMD (ours)** | Mirror descent | Payoff function | Enumerative | Asymptotic |
| APMD | Mirror descent | Gradient field | Enumerative | $O(\log T/\sqrt{T})$ rate |
| DRDA | Discounted FTRL | Policy regularized dynamic | Enumerative | Asymptotic |
| **NashPG (ours)** | Policy gradient | Policy objective | Sample-based | None |
| R-NaD | NeuRD | Reward function | Sample-based | Asymptotic[1] |

Table 1: Comparison with representative multi-round regularization methods.

## 3 Preliminaries

In this paper, we focus on analyzing two-player zero-sum games. We first establish the necessary background and notation for extensive-form games, then present their connection to variational inequality problems and regularized variants that form the foundation of our approach. We use standard mathematical notation for concepts such as L-smoothness, strong convexity, monotonicity, and Bregman divergence (see Appendix A.1 for formal definitions).

---

[1]For R-NaD, the asymptotic exact Nash guarantee refers to the idealized setting where each inner round is solved exactly. This guarantee does not directly transfer to the practical NeuRD implementation with function approximation.

### 3.1 Game Theory Background

We consider two-player zero-sum **extensive-form games** (EFG) with perfect recall and imperfect information. The game is represented by a finite game tree with the following components: Let the players be denoted by $i \in \{1, 2\}$. The set of states is denoted by $\mathcal{S}$, with terminal states $\mathcal{Q} \subseteq \mathcal{S}$ determining payoffs, where player 1's payoff is the negative of player 2's payoff. Each non-terminal state $s \in \mathcal{S} \setminus \mathcal{Q}$ is controlled by either a player or chance. For each player $i$, decision nodes are partitioned into information sets $I \in \mathcal{I}_i$, where the player cannot distinguish between states within the same information set. The set of actions available at information set $I$ is denoted by $A(I)$. A behavioral strategy for player $i$ is a mapping $\pi^{(i)} \colon \mathcal{I}_i \to \Delta(A(I))$, assigning a probability distribution over actions at each information set. A strategy profile $\pi$ is a collection of both players' strategies, and $\pi^{(-i)}$ refers to player $i$'s opponent strategy.

A two-player zero-sum extensive-form game induces an equivalent **normal-form game** (Kuhn, 1953) with payoff matrix $\boldsymbol{A}$, where each row and column correspond to a pure strategy (deterministic action plan across all information sets) for players 1 and 2, respectively. A mixed strategy is a probability distribution over pure strategies. Let $\mathcal{X}$ and $\mathcal{Y}$ denote the mixed strategy spaces for players 1 and 2, with $x \in \mathcal{X}$ and $y \in \mathcal{Y}$ representing their mixed strategies. The payoff function $f(x, y) = x^\top \boldsymbol{A} y$ gives player 1's expected payoff, similarly, player 2's expected payoff is $-f(x, y)$. When referring to mixed strategies in EFGs, we refer to those defined through this normal-form representation.

A **Nash equilibrium** (NE) is a strategy profile $\pi^* = (\pi^{(1)*}, \pi^{(2)*})$ in behavioral strategies or $z^* = (x^*, y^*)$ in mixed strategies, where each player's strategy is a best-response to the other's. Formally, for the normal-form representation, a Nash equilibrium satisfies:

$$x^* \in \arg\max_{x \in \mathcal{X}} f(x, y^*) \quad \text{and} \quad y^* \in \arg\max_{y \in \mathcal{Y}} -f(x^*, y)$$

That is, no player can improve their expected payoff by unilaterally deviating from their equilibrium strategy. Our goal is to develop efficient algorithms to find Nash equilibria in these games.

### 3.2 Connection With Variational Inequality

Finding the Nash equilibrium is equivalent to solving the variational inequality (VI) problem, which forms the foundation for understanding both the challenges and opportunities in our approach.

**Definition 1** (Variational Inequality Problem). *Given a closed convex set $\mathcal{Z} \subseteq \mathbb{R}^n$ and a mapping $G \colon \mathcal{Z} \to \mathbb{R}^n$, the variational inequality problem* $\mathrm{VI}(\mathcal{Z}, G)$ *is to find $z^* \in \mathcal{Z}$ such that*

$$\langle G(z^*), z - z^* \rangle \geq 0 \quad \forall z \in \mathcal{Z}. \tag{2}$$

In a two-player zero-sum extensive-form game, let $\mathcal{X}$ and $\mathcal{Y}$ denote the mixed strategy spaces for players 1 and 2, respectively, and let $\boldsymbol{A}$ be the payoff matrix of the equivalent normal-form game. We recall the fundamental connection between Nash equilibria and variational inequalities by defining $\mathcal{Z} = \mathcal{X} \times \mathcal{Y}$ with strategy profile $z = (x, y) \in \mathcal{Z}$ and the operator $F \colon \mathcal{Z} \to \mathbb{R}^n$:

$$F(z) = \begin{bmatrix} -\nabla_x f(x, y) \\ \nabla_y f(x, y) \end{bmatrix} \tag{3}$$

where $f(x, y) = x^\top \boldsymbol{A} y$ is the payoff function.

**Theorem 2** (Nash-VI Equivalence). *A strategy profile $z^* \in \mathcal{Z}$ is a Nash equilibrium if and only if it solves the variational inequality problem* $\mathrm{VI}(\mathcal{Z}, F)$ *(Facchinei & Pang, 2003, Section 1.4.2):*

$$\langle F(z^*), z - z^* \rangle \geq 0 \quad \forall z \in \mathcal{Z}. \tag{4}$$

Thus, computing Nash equilibrium reduces to solving $\mathrm{VI}(\mathcal{Z}, F)$. However, this creates a significant computational challenge: *The operator $F$ is monotone but not strongly monotone, making* $\mathrm{VI}(\mathcal{Z}, F)$ *difficult to solve directly.*

### 3.3 Regularized Variational Inequality

To address the lack of strong monotonicity in $F$, Sokota et al. (2023) consider a *regularized* VI problem $\text{VI}(\mathcal{Z}, G_\rho)$, where a strongly convex regularizer is added to the operator $F$. The resulting operator $G_\rho$ becomes *strongly monotone*, ensuring a unique solution (Larsson & Patriksson, 1994) and enabling efficient algorithms.

Formally, let $\rho = (\rho_1, \rho_2) \in \mathcal{Z}$ be a reference strategy profile. With a strongly convex function $\psi(z) = \psi_1(x) + \psi_2(y)$, the associated Bregman divergence is

$$B_\psi(z; \rho) = B_{\psi_1}(x; \rho_1) + B_{\psi_2}(y; \rho_2), \tag{5}$$

where $\psi_1$ and $\psi_2$ are strongly convex for each player. The regularized operator $G_\rho : \mathcal{Z} \to \mathbb{R}^n$ is

$$G_\rho(z) = F(z) + \alpha \nabla_z B_\psi(z; \rho), \tag{6}$$

with regularization parameter $\alpha > 0$. The corresponding variational inequality seeks $\hat{z} \in \mathcal{Z}$ such that

$$\langle G_\rho(\hat{z}), z - \hat{z} \rangle \geq 0 \quad \forall z \in \mathcal{Z}. \tag{7}$$

Under Assumption 1, the VI problem $\text{VI}(\mathcal{Z}, G_\rho)$ can be solved efficiently by Magnetic Mirror Descent (MMD) (Sokota et al., 2023):

**Assumption 1** (MMD's Convergence Conditions)**.**

1. *$\psi$ is $\mu$-strongly convex over $\mathcal{Z}$ and differentiable on $\text{int dom}\,\psi$ (the interior of the domain of $\psi$) for some $\mu > 0$.*

2. *$z_{t+1} \in \text{int dom}\,\psi$ for all iterations.*

3. *$F$ is $L$-smooth and $\alpha, \eta$ satisfy $\alpha \geq \mu\eta L^2$.*

**Algorithm 1** (Magnetic Mirror Descent (MMD))**.** *Initialize $z_0 = (x_0, y_0) \in \text{int dom}\,\psi \cap \mathcal{Z}$, reference $\rho = (\rho_1, \rho_2) \in \text{int dom}\,\psi$, and step size $\eta > 0$. At each iteration $t$:*

$$x_{t+1} = \underset{x \in \mathcal{X}}{\arg\max} \left\{ \eta\big(\langle \nabla_{x_t} f(x_t, y_t), x \rangle - \alpha B_{\psi_1}(x; \rho_1)\big) - B_{\psi_1}(x; x_t) \right\},$$

$$y_{t+1} = \underset{y \in \mathcal{Y}}{\arg\min} \left\{ \eta\big(\langle \nabla_{y_t} f(x_t, y_t), y \rangle + \alpha B_{\psi_2}(y; \rho_2)\big) + B_{\psi_2}(y; y_t) \right\}.$$

*With Assumption 1, MMD converges to the unique solution of $\text{VI}(\mathcal{Z}, G_\rho)$.*

## 4 Methodology

To find exact Nash equilibria instead of regularized equilibria, we propose an *iterative-refinement* approach in which we solve a sequence of regularized problems $\text{VI}(\mathcal{Z}, G_\rho)$ and use the solution of the previous step as the reference strategy profile $\rho$ for the next.

Our approach unfolds in two stages. First, we study this iterative-refinement scheme at the mixed-strategy level and instantiate it with Magnetic Mirror Descent, yielding *Iterative MMD* (IMMD) (Sections 4.1–4.2), and prove strictly monotone improvement and convergence to a Nash equilibrium. Next, we derive a practical algorithm, *Nash Policy Gradient* (NASHPG), via a series of approximations (Section 4.3), yielding a method that integrates naturally with RL frameworks.

### 4.1 Regularized VI as an Operator

We begin by viewing the regularized VI problem as an operator that maps a reference strategy profile $\rho$ to the unique solution of $\text{VI}(\mathcal{Z}, G_\rho)$.

**Definition 2** (Regularized VI Operator). *Define the operator $\mathcal{M} : \operatorname{int} \operatorname{dom} \psi \cap \mathcal{Z} \to \operatorname{int} \operatorname{dom} \psi \cap \mathcal{Z}$, where the input is the reference strategy profile $\rho$ and the output $\mathcal{M}(\rho)$ is the unique solution of $\operatorname{VI}(\mathcal{Z}, G_\rho)$. We assume $\mathcal{M}(\rho) \in \operatorname{int} \operatorname{dom} \psi \cap \mathcal{Z}$.*

Our theoretical results show that this operator satisfies three key properties.

First, applying $\mathcal{M}$ never increases the Bregman divergence to a Nash equilibrium. This establishes the soundness of $\mathcal{M}$ as a refinement step.

**Lemma 1** (Distance Non-increase Property). *Given $\alpha > 0$, let $\rho \in \operatorname{int} \operatorname{dom} \psi \cap \mathcal{Z}$ and $z^*$ be any Nash equilibrium (i.e., a solution of $\operatorname{VI}(\mathcal{Z}, F)$). Then:*

$$B_\psi(z^*; \rho) \geq B_\psi(z^*; \mathcal{M}(\rho)) + B_\psi(\mathcal{M}(\rho); \rho). \tag{8}$$

(Proof in Appendix A.4.1.)

Second, the fixed points of $\mathcal{M}$ are precisely the Nash equilibria. In other words, once the procedure reaches a Nash equilibrium, it remains there; this naturally characterizes the termination criterion for our refinement process.

**Lemma 2** (Fixed Point Characterization). *A strategy profile $z^* \in \operatorname{int} \operatorname{dom} \psi \cap \mathcal{Z}$ is a Nash equilibrium if and only if it is a fixed point of the regularized VI operator: $z^* = \mathcal{M}(z^*)$.*

(Proof in Appendix A.4.2.)

Finally, the operator $\mathcal{M}$ is continuous, which will be useful in establishing convergence guarantees in the next section.

**Lemma 3** (Continuity of $\mathcal{M}$). *Assume $\psi$ is $\mu$-strongly convex and continuously differentiable on $\operatorname{int} \operatorname{dom} \psi$ for some $\mu > 0$. Then the operator $\mathcal{M}$ is continuous.*

(Proof in Appendix A.4.3.)

### 4.2 Iterative Refinement toward Equilibrium

Motivated by the above properties, we propose the following procedure: starting from an arbitrary strategy profile, repeatedly apply the operator $\mathcal{M}$ until convergence.

**Algorithm 2** (Iterative $\mathcal{M}$ Method). *Starting with $z_0 \in \operatorname{int} \operatorname{dom} \psi \cap \mathcal{Z}$, at each iteration $t$ compute:*

$$z_{t+1} = \mathcal{M}(z_t) \tag{9}$$

*where we assume $z_t \in \operatorname{int} \operatorname{dom} \psi \cap \mathcal{Z}$ for all iterations.*

The convergence guarantee follows from Lemmas 1 and 2 and the continuity of $\mathcal{M}$ (Lemma 3). We show that each refinement step *strictly* reduces the Bregman divergence to any Nash equilibrium until convergence is reached.

**Theorem 3** (Convergence of Iterative $\mathcal{M}$). *Let $z^*$ be a Nash equilibrium with $z^* \in \operatorname{int} \operatorname{dom} \psi \cap \mathcal{Z}$. Algorithm 2 generates a sequence $\{z_t\}$ such that*

$$B_\psi(z^*; z_t) > B_\psi(z^*; z_{t+1}), \tag{10}$$

*for all $t$ such that $z_t$ is not a Nash equilibrium. Moreover, $\{z_t\}$ converges to a Nash equilibrium, i.e., $\lim_{t \to \infty} z_t$ is a Nash equilibrium.*

(Proof in Appendix A.5.1.)

To implement this iterative method, we employ MMD to solve each regularized VI subproblem. Under Assumption 1 and the existence of a Nash equilibrium $z^* \in \operatorname{int} \operatorname{dom} \psi \cap \mathcal{Z}$, this yields a theoretically well-justified algorithm for finding Nash equilibria in two-player zero-sum games (Algorithm 3). This multi-round structure allows $\alpha$ to be set to large values that easily satisfy the convergence constraint $\alpha \geq \mu \eta L^2$, while still

guaranteeing convergence to a Nash equilibrium. We note that the outer loop of Algorithm 3 is an instance of the classical Bregman proximal-point method for monotone VIs (Eckstein, 1993; Censor et al., 1998) where each outer step implicitly regularizes $F$ with $B_\psi(\cdot; z_t)$, converting $\mathrm{VI}(\mathcal{Z}, F)$ into a strongly-monotone subproblem solved by MMD.

---

**Algorithm 3** Iterative MMD (IMMD)

---

1: Initialize $x_0 \in \operatorname{int} \operatorname{dom} \psi_1 \cap \mathcal{X}$
2: Initialize $y_0 \in \operatorname{int} \operatorname{dom} \psi_2 \cap \mathcal{Y}$
3: Set $\rho_1 \leftarrow x_0$, $\rho_2 \leftarrow y_0$
4: **for** $t = 0, 1, 2, \ldots$ until convergence **do**
5:      Set $x_{t,0} \leftarrow \rho_1$, $y_{t,0} \leftarrow \rho_2$
6:      **for** $k = 0, 1, 2, \ldots$ until convergence **do**
7:          $x_{t,k+1} \leftarrow \arg\max_{x \in \mathcal{X}} \left\{ \eta \left( \langle \nabla_{x_{t,k}} f(x_{t,k}, y_{t,k}), x \rangle - \alpha B_{\psi_1}(x; \rho_1) \right) - B_{\psi_1}(x; x_{t,k}) \right\}$
8:          $y_{t,k+1} \leftarrow \arg\min_{y \in \mathcal{Y}} \left\{ \eta \left( \langle \nabla_{y_{t,k}} f(x_{t,k}, y_{t,k}), y \rangle + \alpha B_{\psi_2}(y; \rho_2) \right) + B_{\psi_2}(y; y_{t,k}) \right\}$
9:      Update $\rho_1 \leftarrow x_{t,K}$, $\rho_2 \leftarrow y_{t,K}$           $\triangleright$ where $K$ is the final inner iteration

---

### 4.3 Practical Implementation: Nash Policy Gradient Algorithm

Although Algorithm 3 provides theoretical guarantees, its reliance on mixed strategies makes it unclear how to mitigate the exponential complexity in the size of the game tree. To address this computational barrier, we develop a practical approximation that preserves the iterative refinement structure while enabling efficient implementation in RL frameworks.

**From Mixed Strategies to Behavioral Strategies.** We illustrate the transformation for player 1; the case for player 2 is symmetric. Consider the mirror-ascent subproblem in line 7 of Algorithm 3. Setting $\psi_1$ to the negative entropy (the canonical choice on the probability simplex) shows that, for sufficiently small step size $\eta$, the mirror step is first-order equivalent to a natural gradient ascent step on the regularized objective (see Appendix A.5 for the derivation):

$$g(x_{t,k}) = f(x_{t,k}, y_{t,k}) - \alpha D_{\mathrm{KL}}(x_{t,k} \,\|\, \rho_1). \tag{11}$$

This mixed-strategy formulation inspired a behavioral strategy analogue. For behavioral strategy profile $\pi = (\pi^{(1)}, \pi^{(2)})$ and reference behavioral strategy profile $\rho = (\rho^{(1)}, \rho^{(2)})$, we design the analogous objective:

$$g(\pi^{(1)}) = \mathbb{E}_{\tau \sim \pi}[R_1(\tau)] - \alpha \, \mathbb{E}_{o \sim \pi} \left[ D_{\mathrm{KL}}\big( \pi^{(1)}(\cdot \mid o) \,\|\, \rho^{(1)}(\cdot \mid o) \big) \right] \tag{12}$$

where $R_1(\tau)$ denotes player 1's payoff along trajectory $\tau$ (a path of actions to a terminal node), $o$ denotes an observation (information set) of player 1, and both trajectories $\tau$ and observations $o$ are sampled under the strategy profile $\pi$. The first term preserves the expected payoff structure, while the regularization term aggregates Kullback–Leibler (KL) divergences across all observations. This design choice for regularization enables efficient estimation within RL frameworks, though it introduces a heuristic element: the expectation over $o$ is taken under the joint policy $\pi$, so the weight placed on each information set depends on the opponent's strategy $\pi^{(2)}$ and on $\pi^{(1)}$ itself through the induced visitation distribution.

**Connection to Partially Observable MDPs.** The shift to behavioral strategies allows a crucial reduction to single-agent RL. When opponent's strategy $\pi^{(-i)}$ is fixed, the game reduces to a Partially Observable Markov Decision Process (POMDP) from player $i$'s perspective (Greenwald et al., 2013). In this view, a behavioral strategy is equivalent to a stochastic policy in a POMDP. Thus, we can leverage any single-agent policy gradient method to optimize the regularized objective in Equation 12. Specifically, the policy and KL regularization gradients can both be estimated via trajectory sampling.

**Practical Algorithm.** These insights lead to our main algorithmic contribution, *Nash Policy Gradient* (NASHPG) presented in Algorithm 4. At each iteration, trajectories are collected under the current joint

policy. Both players then use these trajectories to estimate their policy and KL regularization gradients, and perform regularized policy gradient updates. After $K$ such updates, the reference policies are reset to the updated policies, and this process is repeated for $T$ outer iterations. The parameters $K$ and $T$ can be chosen according to the available computational budget.

A key advantage of NASHPG is its modularity, as it is agnostic to the choice of policy gradient estimator. For instance, when adopting PPO (Schulman et al., 2017), we simply replace the vanilla policy gradient update with PPO's clipped surrogate objective, while leaving the regularization term unchanged. This design allows NASHPG to directly inherit algorithmic advances from the single-agent RL literature.

---

**Algorithm 4** Nash Policy Gradient (NASHPG)

---

1: Initialize policy parameters $\{\theta^{(i)}\}_{i=1}^2$ for policies $\{\pi_{\theta^{(i)}}\}_{i=1}^2$
2: Set reference policies $\rho^{(i)} \leftarrow \pi_{\theta^{(i)}}$ for $i \in \{1, 2\}$
3: **for** $t = 0, 1, \ldots, T - 1$ **do**                              ▷ Outer loop: refining reference policies
4:    **for** $k = 0, 1, \ldots, K - 1$ **do**                         ▷ Inner loop: regularized policy gradient
5:        Sample trajectories $\{\tau\}$ by executing $\pi = (\pi_{\theta^{(1)}}, \pi_{\theta^{(2)}})$ in the environment
6:        **for** each player $i \in \{1, 2\}$ **do**
7:            Estimate policy gradient: $\hat{g}^{(i)} \approx \nabla_{\theta^{(i)}} \mathbb{E}_{\tau \sim \pi}[R_i(\tau)]$
8:            Estimate regularization gradient: $\hat{g}_{\text{reg}}^{(i)} \approx \nabla_{\theta^{(i)}} \mathbb{E}_{o \sim \pi} \big[ D_{\text{KL}}(\pi_{\theta^{(i)}}(\cdot \mid o) \,\|\, \rho^{(i)}(\cdot \mid o)) \big]$
9:            Update player $i$: $\theta^{(i)} \leftarrow \theta^{(i)} + \eta\big(\hat{g}^{(i)} - \alpha \hat{g}_{\text{reg}}^{(i)}\big)$
10:   Update reference policies: $\rho^{(i)} \leftarrow \pi_{\theta^{(i)}}$ for $i \in \{1, 2\}$

---

**Remark.** Unlike IMMD, NASHPG does not carry a formal convergence guarantee. The approximations introduced above (using negative entropy $\psi$ where the Nash equilibria may all lie on the simplex boundary, violating $z^* \in \text{int dom}\, \psi \cap \mathcal{Z}$; replacing the mixed-strategy Bregman divergence with a behavioral-strategy KL term; and using a finite inner loop) break the theoretical chain from Section 4.2. NASHPG is therefore a practical approximation of IMMD that trades formal convergence guarantees for computational tractability.

## 5 Experiments

We organize our empirical evaluation around four questions. **(Q1)** Does NASHPG converge toward a Nash equilibrium in practice? **(Q2)** How sensitive is NASHPG to the regularization strength? **(Q3)** How does NASHPG compare with prior model-free methods as game size increases? **(Q4)** How does the choice of inner update rule affect performance within the same multi-round regularization framework? Complete experimental settings and additional studies are deferred to Appendix B. All code needed to reproduce our experiments is publicly available at `nash_policy_gradient`.

### 5.1 Experimental Setup

**Environments**. We evaluate on seven two-player zero-sum imperfect-information games spanning several orders of magnitude in complexity: Kuhn Poker, Leduc Poker, abrupt Dark Hex $3 \times 3$ (ADH3), abrupt Phantom Tic-Tac-Toe (APTTT), Liar's Dice, Battleship, and Heads-Up No-Limit Texas Hold'em (NLHE). For Kuhn Poker through APTTT, we follow the standard OpenSpiel (Lanctot et al., 2019) rules. For Liar's Dice, we use the standard setting with 5 dice and 6 faces. Battleship follows the standard two-player rules. For Heads-Up NLHE, we use initial stacks of \$200 and a discretized action space with 16 actions per decision point: fold, check/call, 12 raise sizes, and all-in. Table 2 summarizes the scale of these environments; full rules and implementation details are deferred to Appendix B.1.

| Environment | Information states | Evaluation regime |
|---|---|---|
| Kuhn Poker | 12 | Exact exploitability |
| Leduc Poker | 936 | Exact exploitability |
| ADH3 | $\sim 2.733 \times 10^7$ | Exact exploitability |
| APTTT | $\sim 2.331 \times 10^7$ | Exact exploitability |
| Liar's Dice | $\geq 2.9 \times 10^{20}$ | Approx. BR / head-to-head |
| Battleship | $\geq 2.9 \times 10^{35}$ | Approx. BR / head-to-head |
| Heads-Up NLHE | $\geq 1.3 \times 10^{20}$ | Approx. BR / head-to-head |

Table 2: Summary of evaluated environments and their approximate complexity.

**Algorithms**. We compare NASHPG against four model-free baselines: NFSP (Heinrich & Silver, 2016), PSRO (Lanctot et al., 2017), R-NaD (Perolat et al., 2022), and MMD (Sokota et al., 2023). NFSP and PSRO train their best-response policies using PPO. We standardize all shared optimization hyperparameters, including the learning rate, entropy coefficient, and rollout length. To ensure a fair comparison, we also align training budgets across methods wherever applicable. For all environments except Kuhn Poker, we use 10,000 inner-loop updates and 25 outer-loop iterations; for MMD, this corresponds to 250,000 total updates. For Kuhn Poker, we use 2,500 inner-loop updates and 20 outer-loop iterations for all methods. Detailed algorithmic settings are deferred to Appendix B.3.

**Model Architectures**. Within each environment, all algorithms share the same policy and value-network architecture. Depending on the observation structure, the feature extractor can be a multi-layer perceptron, transformer, or convolutional neural network with hidden dimensions ranging from 128 to 512. Full details are provided in Appendix B.4.

**Evaluation**. Our primary metric is exploitability. For Kuhn Poker, Leduc Poker, ADH3, and APTTT, the games are small enough that exploitability can be computed exactly. For Liar's Dice, Battleship, and Heads-Up NLHE, exact exploitability is intractable, we instead report *approximate exploitability*, computed via a PPO best-response agent trained against the learned policy for 10,000 update steps. The BR agent shares the same model architecture as the policy agents, and all algorithms use the same BR training budget for fairness. Because the RL-trained BR is not guaranteed to reach the true best-response value, the resulting approximate exploitability is a *lower bound* on true exploitability; its tightness depends on the quality of the RL training. We therefore complement it with head-to-head matches, where the final policy of each baseline plays against NASHPG and we report the average payoff of NASHPG over 1,024 games. All results are averaged over 5 random seeds and reported as mean $\pm$ standard deviation. Full evaluation details are deferred to Appendix B.4.

**Implementation**. All methods and environments are implemented in JAX (Bradbury et al., 2018). To the best of our knowledge, we provide the first JAX implementations for ADH3, APTTT, Liar's Dice, Battleship, and Heads-Up NLHE, while Kuhn Poker and Leduc Poker are already available in JAX through prior work (Koyamada et al., 2023). Additional implementation details, including the exact exploitability pipeline and solver interfaces, are provided in the Appendix B.

## 5.2 Convergence and Sensitivity to Regularization

We begin by studying how the regularization strength affects practical convergence under a fixed compute budget. Since Kuhn Poker and Leduc Poker are small enough to allow exact exploitability evaluation and fast training, we compare NASHPG, R-NaD, and MMD across a range of regularization strengths. Figure 1 reports final exploitability as a function of the regularization coefficient, while Figure 2 shows the full exploitability trajectories of NASHPG under different regularization strengths.

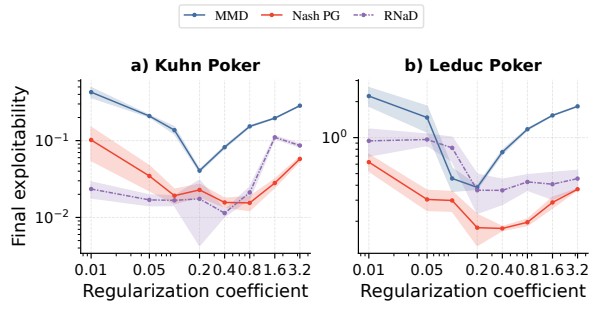

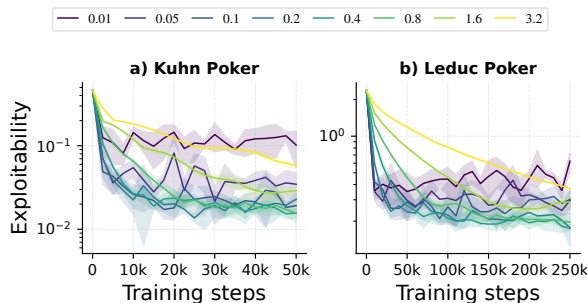

Figure 1: Final exploitability as a function of regularization strength for NASHPG, R-NaD, and MMD on Kuhn Poker and Leduc Poker.

Figure 2: Exploitability trajectories of NASHPG under different regularization strengths.

Across all three methods, performance exhibits a broadly U-shaped dependence on the regularization strength. When the coefficient is too small, the stabilizing effect of regularization is insufficient and stochastic optimization noise degrades convergence. When the coefficient is too large, performance also deteriorates, but for different reasons across methods. For MMD, large regularization moves the solution farther from the Nash equilibrium. For NASHPG and R-NaD, by contrast, large regularization primarily slows refinement, leaving the methods stable, but converge more slowly within a fixed compute budget. Figure 2 illustrates this tradeoff for NASHPG and shows that NASHPG exhibits no sign of divergence at $\alpha \geq 0.2$ in both games. Moreover, Figure 1 shows that $\alpha = 0.2$ performs well across both games for all three methods, we therefore use $\alpha = 0.2$ in all subsequent experiments.

## 5.3 Comparison Across Game Sizes

We next evaluate how NASHPG compares with prior model-free methods as game size increases. Across all environments considered, we do not observe divergence in NASHPG, which is encouraging evidence for its practical stability.

Figure 3 reports exploitability on the smaller benchmark games where exact evaluation is available, and Figure 4 reports approximate exploitability on the larger domains. In the exact-evaluation games, NASHPG consistently reaches low exploitability and is competitive with, or superior to, the other model-free methods. The only exception is Kuhn Poker, where its performance is comparable rather than clearly better.

Population-based methods, NFSP and PSRO, often converge more slowly than NASHPG, and in some environments fail to match the final performance of the regularization-based methods. This is unsurprising, since population-based methods must enumerate all pure strategies in the worst case, which grow exponentially in the size of the game (Lanctot et al., 2017).

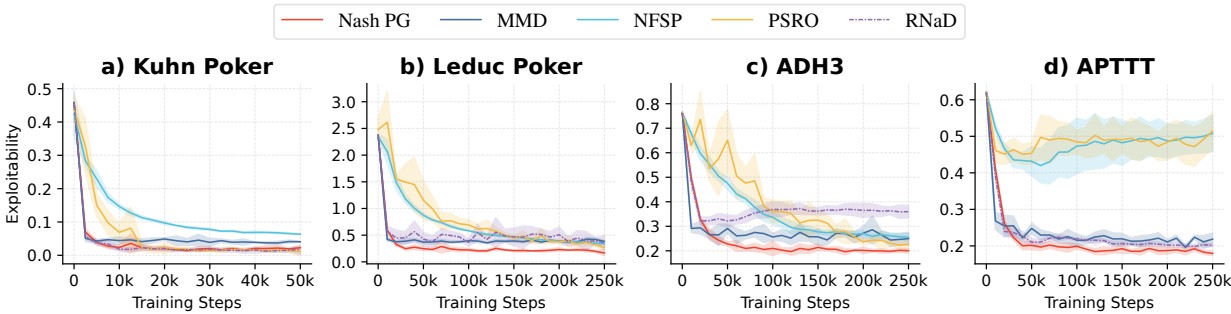

Figure 3: Exploitability on games where exact evaluation is available. Error bars denote standard deviation over 5 seeds.

Relative to MMD, NASHPG typically achieves lower final exploitability in our experiments, which is consistent with the interpretation of NASHPG as an iterative refinement procedure that progressively removes the bias induced by a fixed regularization reference. R-NaD often performs comparably to NASHPG early in training but can later become trapped in a suboptimal regime, which motivates the controlled study in the next subsection (Section 5.4).

| Environment | vs MMD | vs NFSP | vs PSRO | vs RNaD |
|---|---|---|---|---|
| Liar's Dice | $+0.008 \pm 0.035^{\dagger}$ | $+0.119 \pm 0.050$ | $+0.153 \pm 0.046$ | $+0.163 \pm 0.041$ |
| Heads-Up Poker | $+0.111 \pm 0.023$ | $+0.163 \pm 0.012$ | $+0.149 \pm 0.015$ | $+0.104 \pm 0.064$ |
| Battleship | $+0.004 \pm 0.007^{\dagger}$ | $+0.174 \pm 0.017$ | $+0.092 \pm 0.008$ | $+0.107 \pm 0.008$ |

Table 3: Head-to-head average payoff of NASHPG against each baseline ($\pm$ std over 5 runs). Positive values indicate NASHPG wins. $^{\dagger}$Cell is within one standard deviation of zero.

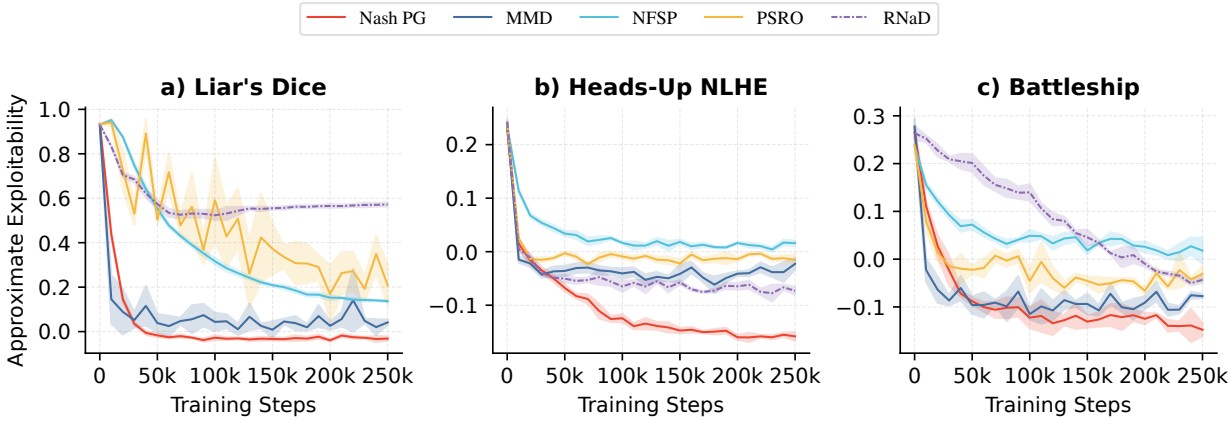

Figure 4: Approximate exploitability in large games. Error bars denote standard deviation over 5 seeds.

In the large games, exploitability is only approximate by training PPO best-response agents. We therefore complement approximate exploitability with head-to-head matches against NASHPG to measure relative strength of each baseline. Table 3 shows that NASHPG achieves positive average payoff against all competing methods across these domains. Two cells (Liar's Dice and Battleship vs. MMD, marked $^{\dagger}$) fall within one standard deviation of zero, indicating the gap is not statistically reliable; NASHPG is nonetheless at least comparable to MMD in these settings. Moreover, the ordering induced by the head-to-head results broadly agrees with the ordering suggested by Figure 4, which provides some evidence that approximate exploitability evaluation remains informative.

Overall, these results suggest that NASHPG remains stable across a wide range of game sizes and offers a favorable combination of convergence speed and final performance.

## 5.4 Does a More Mature Inner Update Rule Improve Multi-Round Regularization?

Our introduction (Section 1) hypothesized that the weak empirical performance of R-NaD may stem from its inner optimization rule rather than from the regularization framework itself. In particular, R-NaD relies on NeuRD (Hennes et al., 2020), whereas NASHPG uses PPO-style policy optimization, which benefits from a substantially more mature set of algorithmic stabilizations. To test this hypothesis, we construct a controlled variant of R-NaD in which we keep the reward transformation and outer-loop regularization structure of R-NaD, but replace NeuRD with PPO as the inner optimizer. We denote this variant by RNaD-PPO.

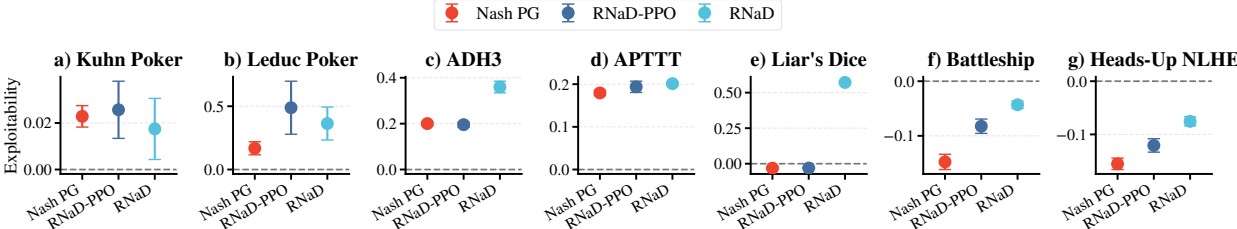

Figure 5: Effect of replacing NeuRD with PPO within the R-NaD framework.

Figure 5 reveals a more nuanced pattern. In the smallest games, including Kuhn Poker and Leduc Poker, replacing NeuRD with PPO can hurt performance. In this regime, optimization is already sufficiently stable that the theoretical structure of the update appears to matter more than additional reinforcement-learning heuristics. In contrast, as the environments become larger and optimization becomes more challenging, RNaD-PPO substantially improves over R-NaD. Replacing NeuRD with PPO dramatically narrows the gap between R-NaD and NASHPG, and in some cases yields performance comparable to NASHPG. This provides evidence that, in larger and more challenging games, the practical bottleneck in R-NaD lies in its inner update rule rather than in the regularization framework itself, and that the performance gap in these settings is primarily attributable to using NeuRD rather than PPO as the inner optimizer.

### 5.5 Discussion

Taken together, the results offer encouraging evidence on all four questions. First, NASHPG exhibits stable practical convergence when the regularization strength is chosen appropriately. Second, it compares favorably with prior model-free methods across a broad range of game sizes. Third, the controlled ablation on R-NaD supports our central hypothesis: the practical success of sample-based multi-round regularization depends crucially on the inner policy update rule. Additionally, NASHPG is straightforward to implement, our JAX implementation of NASHPG differs from Independent PPO (Albrecht et al., 2024; Schulman et al., 2017) implementation by only about 20 lines of code, making it easy to adopt within existing codebases.

## 6 Conclusion and Future Work

We proposed IMMD, a regularization-based framework that achieves convergence to Nash equilibria in two-player zero-sum imperfect-information games by iteratively refining the reference policy. We proved strictly monotonic improvement of the solution over iterations, and established convergence guarantees to Nash equilibria. Inspired by these theoretical findings, we developed NASHPG, a practical algorithm that closely mirrors standard policy gradient methods while adding a KL regularization term and periodically updated reference policy. Empirically, NASHPG achieves low exploitability and scales effectively to large domains such as *Battleship* and *Heads-Up No-Limit Texas Hold'em*. A central appeal of NASHPG is its accessibility, as the method requires only minor modifications to standard policy gradient algorithms, lowering the barrier to entry for practitioners in MARL.

Several directions remain open for future work. On the theoretical side, while IMMD is fully grounded by our analysis, some steps used to derive the practical NASHPG objective from the mixed-strategy formulation remain heuristic (Section 4.3) and would benefit from a more principled justification. A further theoretical direction is to strengthen the convergence result for IMMD to a finite-time bound; we hypothesize that the regularization strength $\alpha$ will appear naturally in such a bound, which could provide a principled explanation for the U-shaped dependence on $\alpha$ observed empirically in Figure 1. Another natural direction is to extend the framework beyond 2p0s games to general games settings, where equilibrium selection becomes an additional challenge (Christianos et al., 2023). Improving the sample efficiency of self-play training is another open direction; approaches based on intrinsic rewards (Schäfer et al., 2022) could accelerate convergence. On the empirical side, benchmarking NASHPG against domain-specific state-of-the-art methods, such as Pluribus (Brown & Sandholm, 2019), in large poker settings would further clarify its strengths. It

would also be valuable to evaluate NASHPG more broadly in additional domains, including continuous-action games and other high-dimensional environments, to better understand its practical range.

**Broader Impact Statement**

This work develops theoretical and algorithmic advances for finding Nash equilibria in two-player zero-sum imperfect-information games. The primary applications are in game-solving and multi-agent reinforcement learning research. We do not foresee direct negative societal impacts beyond those associated with general advances in AI.

**Author Contributions**

**Acknowledgments**

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

# A    Theoretical Analysis

In this appendix, we provide detailed proofs for the main theoretical results presented in Section 4.1 and Section 4.2. We begin by establishing the mathematical foundation and recall the problem setup.

## A.1    Mathematical Notation

For a function $f$, we denote its domain by $\mathrm{dom}\, f$ and for a set $\mathcal{D}$, we write $\mathrm{int}\,\mathcal{D}$ for its interior.

**Definition 3** ($L$-smooth function). *A differentiable function $f : \mathcal{Z} \to \mathbb{R}$ is $L$-smooth on $\mathcal{Z} \subseteq \mathbb{R}^n$ if its gradient is $L$-Lipschitz continuous:*

$$\|\nabla f(z) - \nabla f(z')\| \leq L\|z - z'\| \quad \forall z, z' \in \mathcal{Z}.$$

**Definition 4** ($\mu$-strong convex function). *A differentiable function $f : \mathcal{Z} \to \mathbb{R}$ is $\mu$-strongly convex on $\mathcal{Z}$ if there exists $\mu > 0$ such that*

$$f(z') \geq f(z) + \langle \nabla f(z), z' - z \rangle + \tfrac{\mu}{2}\|z' - z\|^2 \quad \forall z, z' \in \mathcal{Z}.$$

**Definition 5** (Monotone operator). *An operator $G : \mathcal{Z} \to \mathbb{R}^n$ is monotone on $\mathcal{Z}$ if*

$$\langle G(z) - G(z'), z - z' \rangle \geq 0 \quad \forall z, z' \in \mathcal{Z}.$$

*It is $\mu$-strongly monotone on $\mathcal{Z}$ if there exists $\mu > 0$ such that*

$$\langle G(z) - G(z'), z - z' \rangle \geq \mu\|z - z'\|^2 \quad \forall z, z' \in \mathcal{Z}.$$

**Definition 6** (Bregman divergence). *Let $\psi : \mathcal{Z} \to \mathbb{R}$ be a differentiable, strongly convex function. For $z \in \mathrm{dom}\,\psi$ and $z' \in \mathrm{int}\,\mathrm{dom}\,\psi$, the Bregman divergence is defined as*

$$B_\psi(z; z') = \psi(z) - \psi(z') - \langle \nabla\psi(z'), z - z' \rangle,$$

*which satisfies $B_\psi(z; z') \geq 0$.*

## A.2    Problem Setup

**Two-Player Zero-Sum Games.**    We consider two-player zero-sum extensive-form games with perfect recall. Their equivalent normal-form representation is given by a payoff matrix $\boldsymbol{A}$, where rows and columns correspond to the pure strategies of players 1 and 2. Let $\mathcal{X}$ and $\mathcal{Y}$ denote the mixed strategy spaces (probability simplices) of the two players, and define the strategy profile space $\mathcal{Z} = \mathcal{X} \times \mathcal{Y}$. A strategy profile is $z = (x, y) \in \mathcal{Z}$, and the payoff to player 1 is

$$f(x, y) = x^\top \boldsymbol{A} y.$$

**Nash Equilibrium.**    A strategy profile $z^* = (x^*, y^*) \in \mathcal{Z}$ is a Nash equilibrium if neither player can improve unilaterally:

$$x^* \in \arg\max_{x \in \mathcal{X}} f(x, y^*) \quad \text{and} \quad y^* \in \arg\max_{y \in \mathcal{Y}} -f(x^*, y).$$

**Variational Inequality Formulation.**    The Nash equilibrium problem can be equivalently expressed as a variational inequality (VI). Define the operator $F : \mathcal{Z} \to \mathbb{R}^n$ by

$$F(z) = \begin{bmatrix} -\nabla_x f(x, y) \\ \nabla_y f(x, y) \end{bmatrix} = \begin{bmatrix} -\boldsymbol{A}y \\ \boldsymbol{A}^\top x \end{bmatrix}. \tag{13}$$

Then $z^*$ is a Nash equilibrium if and only if it solves $\mathrm{VI}(\mathcal{Z}, F)$:

$$\langle F(z^*), z - z^* \rangle \geq 0 \quad \forall z \in \mathcal{Z}.$$

**Regularized Variational Inequality.** With a strongly convex regularizer $\psi(z) = \psi_1(x) + \psi_2(y)$ with associated Bregman divergence

$$B_\psi(z; \rho) = B_{\psi_1}(x; \rho_1) + B_{\psi_2}(y; \rho_2),$$

where $\rho = (\rho_1, \rho_2) \in \mathcal{Z}$ is a reference strategy profile. The regularized operator is defined as

$$G_\rho(z) = F(z) + \alpha \nabla_z B_\psi(z; \rho) \quad \alpha > 0, \tag{14}$$

or equivalently

$$G_\rho(z) = F(z) + \alpha\big(\nabla\psi(z) - \nabla\psi(\rho)\big) \quad \alpha > 0. \tag{15}$$

The corresponding regularized VI seeks $\hat{z} \in \mathcal{Z}$ such that

$$\langle G_\rho(\hat{z}), z - \hat{z}\rangle \geq 0 \quad \forall z \in \mathcal{Z}.$$

**Regularized VI Operator.** The central object of our analysis is the mapping

$$\mathcal{M} : \operatorname{int} \operatorname{dom} \psi \cap \mathcal{Z} \rightarrow \operatorname{int} \operatorname{dom} \psi \cap \mathcal{Z},$$

which takes a reference profile $\rho$ to the unique solution $\mathcal{M}(\rho)$ of $\mathrm{VI}(\mathcal{Z}, G_\rho)$. Formally, $\mathcal{M}(\rho)$ is the unique point satisfying

$$\langle G_\rho(\mathcal{M}(\rho)), z - \mathcal{M}(\rho)\rangle \geq 0 \quad \forall z \in \mathcal{Z}.$$

### A.3 Preliminary Results

**Lemma 4** (Three-Point Property). *For any strongly convex function $\psi$ and points $a \in \operatorname{dom} \psi$ and $\{b, c\} \subset \operatorname{int} \operatorname{dom} \psi$, the Bregman divergence satisfies:*

$$B_\psi(a; b) = B_\psi(a; c) + B_\psi(c; b) + \langle \nabla\psi(c) - \nabla\psi(b), a - c\rangle \tag{16}$$

*Proof.* By definition of Bregman divergence:

$$B_\psi(a; b) = \psi(a) - \psi(b) - \langle \nabla\psi(b), a - b\rangle \tag{17}$$
$$B_\psi(a; c) = \psi(a) - \psi(c) - \langle \nabla\psi(c), a - c\rangle \tag{18}$$
$$B_\psi(c; b) = \psi(c) - \psi(b) - \langle \nabla\psi(b), c - b\rangle \tag{19}$$

Direct algebraic manipulation yields the result. $\qquad\square$

**Lemma 5** (Monotonicity of Operator $F$). *The operator $F : \mathcal{Z} \rightarrow \mathbb{R}^n$ defined in the problem setup is monotone on $\mathcal{Z}$.*

*Proof.* Let $z_1 = (x_1, y_1)$ and $z_2 = (x_2, y_2)$ be two strategy profiles in $\mathcal{Z}$. We have:

$$\langle F(z_1) - F(z_2), z_1 - z_2\rangle \tag{20}$$
$$= \langle -\boldsymbol{A}y_1 + \boldsymbol{A}y_2, x_1 - x_2\rangle + \langle \boldsymbol{A}^\top x_1 - \boldsymbol{A}^\top x_2, y_1 - y_2\rangle \tag{21}$$
$$= -(x_1 - x_2)^\top \boldsymbol{A}(y_1 - y_2) + (x_1 - x_2)^\top \boldsymbol{A}(y_1 - y_2) \tag{22}$$
$$= 0 \geq 0 \tag{23}$$

where we used the definition $F(z) = [-\boldsymbol{A}y; \boldsymbol{A}^\top x]$ from the problem setup. $\qquad\square$

**Lemma 6** (Strong Monotonicity of $G_\rho$). *Let $\psi$ be $\mu$-strongly convex on $\operatorname{int} \operatorname{dom} \psi$ with $\mu > 0$, then $G_\rho$ is $\alpha\mu$-strongly monotone on $\mathcal{Z}$, i.e.,*

$$\langle G_\rho(u) - G_\rho(v), u - v\rangle \geq \alpha\mu\|u - v\|^2 \quad \text{for all } u, v \in \mathcal{Z}.$$

*Proof.* For any $u, v \in \mathcal{Z}$,

$$\langle G_\rho(u) - G_\rho(v), u - v\rangle = \langle F(u) - F(v), u - v\rangle + \alpha\langle \nabla\psi(u) - \nabla\psi(v), u - v\rangle.$$

By Lemma 5, $F$ is monotone, so the first term is nonnegative. By $\mu$-strong convexity of $\psi$, the second term is at least $\alpha\mu\|u - v\|^2$. This proves $\alpha\mu$-strong monotonicity of $G_\rho$. $\qquad\square$

### A.4 Main Theoretical Results

#### A.4.1 Proof of Lemma 1

**Lemma 1** (Distance Non-increase Property). *Given $\alpha > 0$, let $\rho \in \text{int dom}\,\psi \cap \mathcal{Z}$ and $z^*$ be any Nash equilibrium (i.e., a solution of $\text{VI}(\mathcal{Z}, F)$). Then:*

$$B_\psi(z^*; \rho) \geq B_\psi(z^*; \mathcal{M}(\rho)) + B_\psi(\mathcal{M}(\rho); \rho). \tag{8}$$

*Proof.* For notational convenience, define $z_\rho := \mathcal{M}(\rho)$. The Nash equilibrium $z^*$ satisfies the VI optimality condition for $F$:

$$\langle F(z^*), z - z^* \rangle \geq 0, \quad \forall z \in \mathcal{Z}. \tag{24}$$

Similarly, $z_\rho$ satisfies the VI optimality condition for the regularized operator $G_\rho$:

$$\langle G_\rho(z_\rho), z - z_\rho \rangle \geq 0, \quad \forall z \in \mathcal{Z}. \tag{25}$$

Substitute $z = z_\rho$ in equation 24 and $z = z^*$ in equation 25, then add the two inequalities:

$$\langle F(z^*), z_\rho - z^* \rangle + \langle F(z_\rho) + \alpha \nabla_{z_\rho} B_\psi(z_\rho; \rho), z^* - z_\rho \rangle \geq 0. \tag{26}$$

Rearranging equation 26 gives

$$\alpha \langle \nabla_{z_\rho} B_\psi(z_\rho; \rho), z^* - z_\rho \rangle \geq \langle F(z^*) - F(z_\rho), z^* - z_\rho \rangle. \tag{27}$$

By Lemma 5, $F$ is monotone on $\mathcal{Z}$, so the right-hand side of equation 27 is nonnegative. Hence, since $\alpha > 0$, we obtain

$$\langle \nabla_{z_\rho} B_\psi(z_\rho; \rho), z^* - z_\rho \rangle \geq 0. \tag{28}$$

And since $\nabla_{z_\rho} B_\psi(z_\rho; \rho) = \nabla \psi(z_\rho) - \nabla \psi(\rho)$, equation 28 becomes

$$\langle \nabla \psi(z_\rho) - \nabla \psi(\rho), z^* - z_\rho \rangle \geq 0. \tag{29}$$

Finally, applying the three-point property of Bregman divergences (Lemma 4) with $a = z^*$, $c = z_\rho$, and $b = \rho$, we have

$$B_\psi(z^*; \rho) = B_\psi(z^*; z_\rho) + B_\psi(z_\rho; \rho) + \langle \nabla \psi(z_\rho) - \nabla \psi(\rho), z^* - z_\rho \rangle. \tag{30}$$

Combining equation 30 with equation 29 immediately gives

$$B_\psi(z^*; \rho) \geq B_\psi(z^*; z_\rho) + B_\psi(z_\rho; \rho).$$

$\square$

#### A.4.2 Proof of Lemma 2

**Lemma 2** (Fixed Point Characterization). *A strategy profile $z^* \in \text{int dom}\,\psi \cap \mathcal{Z}$ is a Nash equilibrium if and only if it is a fixed point of the regularized VI operator: $z^* = \mathcal{M}(z^*)$.*

*Proof.* ($\Rightarrow$) Suppose $z^*$ is a Nash equilibrium. By definition, $z^*$ solves $\text{VI}(\mathcal{Z}, F)$:

$$\langle F(z^*), z - z^* \rangle \geq 0, \quad \forall z \in \mathcal{Z}. \tag{31}$$

Consider the regularized operator $G_{z^*}(z)$. By the VI optimality condition, $z^*$ also solves $\text{VI}(\mathcal{Z}, G_{z^*})$, since

$$\langle G_{z^*}(z^*), z - z^* \rangle = \langle F(z^*), z - z^* \rangle + \alpha \langle \nabla_z B_\psi(z^*; z^*), z - z^* \rangle = \langle F(z^*), z - z^* \rangle \geq 0,$$

using $\nabla_z B_\psi(z^*; z^*) = 0$. By definition of the operator $\mathcal{M}$, the unique solution of $\text{VI}(\mathcal{Z}, G_{z^*})$ is $\mathcal{M}(z^*)$. Therefore, $z^* = \mathcal{M}(z^*)$.

($\Leftarrow$) Conversely, suppose $z^* = \mathcal{M}(z^*)$. By definition of operator $\mathcal{M}$, $z^*$ solves VI$(\mathcal{Z}, G_{z^*})$:

$$\langle G_{z^*}(z^*), z - z^* \rangle \geq 0, \quad \forall z \in \mathcal{Z}. \tag{32}$$

Expanding $G_{z^*}(z^*) = F(z^*) + \alpha \nabla_z B_\psi(z^*; z^*) = F(z^*)$, we see that $z^*$ satisfies

$$\langle F(z^*), z - z^* \rangle \geq 0, \quad \forall z \in \mathcal{Z},$$

i.e., $z^*$ is a Nash equilibrium. $\qquad\square$

### A.4.3 Proof of Lemma 3

**Lemma 3** (Continuity of $\mathcal{M}$). *Assume $\psi$ is $\mu$-strongly convex and continuously differentiable on* int dom $\psi$ *for some $\mu > 0$. Then the operator $\mathcal{M}$ is continuous.*

*Proof.* Fix $\rho \in \text{int dom}\,\psi \cap \mathcal{Z}$, let $\{\rho_n\} \subset \text{int dom}\,\psi \cap \mathcal{Z}$ be any sequence with $\rho_n \to \rho$, and for notational convenience we write

$$z_n := \mathcal{M}(\rho_n), \qquad z_\rho := \mathcal{M}(\rho).$$

By the VI optimality conditions for $z_n$ and $z_\rho$ we have

$$\langle G_{\rho_n}(z_n), z_\rho - z_n \rangle \geq 0 \quad \text{and} \quad \langle G_\rho(z_\rho), z_n - z_\rho \rangle \geq 0.$$

Adding these two inequalities yields

$$\langle G_{\rho_n}(z_n) - G_\rho(z_\rho), \, z_\rho - z_n \rangle \geq 0,$$

rearranges to get

$$\langle G_{\rho_n}(z_n) - G_{\rho_n}(z_\rho), \, z_n - z_\rho \rangle \leq \langle G_{\rho_n}(z_\rho) - G_\rho(z_\rho), \, z_\rho - z_n \rangle. \tag{33}$$

By $\alpha\mu$-strong monotonicity of $G_{\rho_n}$ (Lemma 6), the left-hand side of equation 33 is bounded below by $\alpha\mu\|z_n - z_\rho\|^2$. Applying the Cauchy–Schwarz inequality to the right-hand side of equation 33 yields

$$\alpha\mu\|z_n - z_\rho\|^2 \leq \|G_{\rho_n}(z_\rho) - G_\rho(z_\rho)\| \cdot \|z_n - z_\rho\|.$$

If $z_n = z_\rho$ the desired convergence holds trivially. Otherwise we divide both sides by $\|z_n - z_\rho\| > 0$ to obtain

$$\|z_n - z_\rho\| \leq \frac{1}{\alpha\mu} \|G_{\rho_n}(z_\rho) - G_\rho(z_\rho)\|. \tag{34}$$

Since

$$G_{\rho_n}(z_\rho) - G_\rho(z_\rho) = \alpha\big(\nabla\psi(\rho) - \nabla\psi(\rho_n)\big),$$

we have that equation 34 simplifies to

$$\|z_n - z_\rho\| \leq \frac{1}{\mu} \|\nabla\psi(\rho_n) - \nabla\psi(\rho)\|. \tag{35}$$

Since $\nabla\psi$ is continuous on int dom $\psi$ and $\rho_n \to \rho$, the right-hand side of equation 35 vanishes as $n \to \infty$. Therefore $\mathcal{M}(\rho_n) \to \mathcal{M}(\rho)$. As this holds for every sequence $\rho_n \to \rho$, the mapping $\mathcal{M}$ is continuous on int dom $\psi \cap \mathcal{Z}$. $\qquad\square$

### A.5 First-Order Equivalence of the Mirror Step (Equation 11)

**Proposition 1** (First-Order Equivalence). *With $\psi_1$ set to the negative entropy, the mirror-ascent step for player 1 (line 7 of Algorithm 3) satisfies*

$$x_{k+1} = x_k + \eta \, \nabla_x^{\text{nat}} g(x_k) + O(\eta^2),$$

*where $g(x) = f(x, y_k) - \alpha \, \text{KL}(x \| \rho_1)$ and the natural gradient on the simplex is $\nabla_{x_i}^{\text{nat}} g(x) := x_i \big[ \nabla_{x_i} g(x) - \langle x, \nabla_x g(x) \rangle \big]$.*

*Proof.* Since $f(x, y_k) = x^\top \boldsymbol{A} y_k$ is linear in $x$, we have $\langle \nabla_x f(x_k, y_k), x \rangle = f(x, y_k)$ exactly (the gradient $\boldsymbol{A} y_k$ is independent of $x_k$). Hence the mirror step is

$$x_{k+1} = \operatorname{argmax}_{x \in \mathcal{X}} \left\{ \eta\, g(x) - \mathrm{KL}(x \| x_k) \right\}. \tag{36}$$

This is exact, not a linearization. The KKT condition at the interior maximizer $x_{k+1}$ (with multiplier $c$ for $\sum_i x_i = 1$) reads

$$\eta\, \nabla_{x_i} g(x_{k+1}) - \log(x_{k+1,i}/x_{k,i}) = c \quad \forall i.$$

At $\eta = 0$ the unique solution is $x_{k+1} = x_k$ with $c = 0$. Differentiating implicitly in $\eta$ at $\eta = 0$ and writing $\dot{x} = \frac{d}{d\eta} x_{k+1}\big|_{\eta=0}$:

$$\nabla_{x_i} g(x_k) - \dot{x}_i / x_{k,i} = \dot{c} \quad \forall i.$$

Multiplying by $x_{k,i}$ and summing over $i$, with $\sum_i \dot{x}_i = 0$, gives $\dot{c} = \langle x_k, \nabla_x g(x_k) \rangle$. Substituting back:

$$\dot{x}_i = x_{k,i} \big[ \nabla_{x_i} g(x_k) - \langle x_k, \nabla_x g(x_k) \rangle \big] = \nabla^{\mathrm{nat}}_{x_i} g(x_k). \qquad \square$$

### A.5.1 Proof of Convergence of Iterative Refinement Procedure

**Theorem 3** (Convergence of Iterative $\mathcal{M}$). *Let $z^*$ be a Nash equilibrium with $z^* \in \operatorname{int} \operatorname{dom} \psi \cap \mathcal{Z}$. Algorithm 2 generates a sequence $\{z_t\}$ such that*

$$B_\psi(z^*; z_t) > B_\psi(z^*; z_{t+1}), \tag{10}$$

*for all $t$ such that $z_t$ is not a Nash equilibrium. Moreover, $\{z_t\}$ converges to a Nash equilibrium, i.e., $\lim_{t \to \infty} z_t$ is a Nash equilibrium.*

*Proof.* Fix any Nash equilibrium $z^*$. For the first part of the theorem, we show the strict inequality holds for all iterations before convergence occurs. Apply Lemma 1 with $\rho = z_t$ and recall $z_{t+1} = \mathcal{M}(z_t)$ to obtain

$$B_\psi(z^*; z_t) \geq B_\psi(z^*; z_{t+1}) + B_\psi(z_{t+1}; z_t).$$

If the iteration has not converged then $z_{t+1} \neq z_t$, therefore $B_\psi(z_{t+1}; z_t) > 0$. Hence the preceding inequality is strict, yielding

$$B_\psi(z^*; z_t) > B_\psi(z^*; z_{t+1}).$$

Next, we show the sequence converges to a Nash equilibrium. Iterating the inequality from Lemma 1, we get, for any $k \geq 1$,

$$\begin{aligned} B_\psi(z^*; z_0) &\geq B_\psi(z^*; z_1) + B_\psi(z_1; z_0) \\ &\geq B_\psi(z^*; z_2) + B_\psi(z_2; z_1) + B_\psi(z_1; z_0) \\ &\geq B_\psi(z^*; z_k) + \sum_{\ell=0}^{k-1} B_\psi(z_{\ell+1}; z_\ell) \,. \end{aligned} \tag{37}$$

As all terms in the RHS are non-negative, this implies that the partial sums of the series of non-negative terms $\sum_{\ell=0}^{k-1} B_\psi(z_{\ell+1}; z_\ell)$ are uniformly bounded, and so the series converges. In particular $B_\psi(z_{k+1}; z_k) \to 0$, which we will use below. Moreover, Equation equation 37 also implies $B_\psi(z^*; z_k) \leq B_\psi(z^*; z_0)$, and the sequence $\{z_k\}_k$ belongs to the sublevel set $S \coloneqq \{z : B_\psi(z^*; z) \leq r\}$ where $r \coloneqq B_\psi(z^*; z_0)$. Since $z^* \in \operatorname{int} \operatorname{dom} \psi$, the function $z \mapsto B_\psi(z^*; z)$ diverges to $+\infty$ as $z$ approaches $\partial \operatorname{dom} \psi$; hence $S$ is compact and contained in $\operatorname{int} \operatorname{dom} \psi$. As a result, there exists a convergent subsequence $\{z_{\phi(k)}\}_k$ (where $\phi \colon \mathbb{N} \to \mathbb{N}$ is increasing) with limit

$$z_\infty \coloneqq \lim_{k \to \infty} z_{\phi(k)} \,.$$

Since $S$ is closed and $\{z_{\phi(k)}\} \subset S$, we have $z_\infty \in S \subset \operatorname{int} \operatorname{dom} \psi$. We show that $z_\infty$ is a Nash equilibrium. From the above,

$$B_\psi(\mathcal{M}(z_{\phi(k)}); z_{\phi(k)}) = B_\psi(z_{\phi(k)+1}; z_{\phi(k)}) \xrightarrow[k \to \infty]{} 0$$

while, by continuity of the Bregman divergence and of the $\mathcal{M}$ operator (Lemma 3),

$$B_\psi(\mathcal{M}(z_{\phi(k)}); z_{\phi(k)}) \xrightarrow[k \to \infty]{} B_\psi(\mathcal{M}(z_\infty); z_\infty)$$

and so, by uniqueness of the limit, $B_\psi(\mathcal{M}(z_\infty); z_\infty) = 0$. Lemma 2 then allows us to conclude that $z_\infty$ is a Nash equilibrium.

We can then conclude that $z_t$ converges to $z_\infty$: indeed, applying Lemma 1 to the Nash equilibrium $z_\infty$, we get as the beginning of the proof that, for all $t, s \geq 0$,

$$B_\psi(z_\infty; z_t) \geq B_\psi(z_\infty; z_{t+s})$$

(i.e., $B_\psi(z_\infty; z_t)$ is a non-increasing, non-negative sequence). Since $\lim_{k \to \infty} z_{\phi(k)} = z_\infty$, this easily allows us to conclude that the sequence $\{z_t\}_t$ converges to $z_\infty$ as well. (In more detail: fixing $\varepsilon > 0$, there exists $k_\varepsilon \geq 0$ such that, for all $k \geq k_\varepsilon$, $B_\psi(z_\infty; z_{\phi(k)}) \leq \varepsilon$. Then, for all $t \geq \phi(k_\varepsilon)$, we have $B_\psi(z_\infty; z_t) \leq B_\psi(z_\infty; z_{\phi(k_\varepsilon)}) \leq \varepsilon$.) This concludes the proof. $\qquad\square$

# B   Experiment Details

## B.1   Environment Domain

We evaluate NashPG across seven 2p0s IIGs. Our benchmark covers **Kuhn Poker**, **Leduc Poker**, **Abrupt Dark Hex (3×3)**, **Abrupt Phantom Tic-Tac-Toe**, **Liar's Dice**, **Battleship**, and **Heads-Up No-Limit Texas Hold'em**. For the two imperfect-information board games (Dark Hex and Phantom Tic-Tac-Toe), we apply the *abrupt* turn rule, i.e. any blocked move immediately ends the acting player's turn, and we follow the configurations provided by OpenSpiel (Lanctot et al., 2019). Architectural details are summarised in Table 5. We denote by $\Delta$ the true gain or loss within a game or hand in poker-related environments. For instance, in Heads-Up No-Limit Texas Hold'em, if two players build a \$50 pot through betting and calling, and Player 1 bets an additional \$30 and Player 2 folds, then $\Delta = +25$ for Player 1 and $-25$ for Player 2. Comprehensive environment rules, architectural specifications, and reward formulations are listed below for reproducibility.

**Kuhn Poker**: A three-card, two-player imperfect-information poker game in which both players ante one chip and act sequentially by betting or passing. Rewards are zero during play; at termination, the winner receives the pot and the loser forfeits their contribution. Observations include three dimensions for one-hot encoding of the private card (J=0, Q=1, K=2) and four dimensions for the complete betting history from the current player's perspective.

**Leduc Poker**: An variant of Poker with six cards (two suits, three ranks), two betting rounds, and a public card revealed in the second round; raises are of fixed size. Rewards are zero during play; a fold immediately ends the hand in favour of the non-folding player, otherwise showdown is decided by pair strength or high card. Observations include 14 dimensions for one-hot encoding of the public and private cards (J♠=0, Q♠=1, K♠=2, J♡=3, Q♡=4, K♡=5), one for the revealed public-card indicator, two for the round indicator, two for player position, and 32 for the full betting history.

**Abrupt Dark Hex (3×3)**: A two-player imperfect-information variant of Hex on a $3 \times 3$ board: Player 0 aims to connect top to bottom, and Player 1 aims to connect left to right. Each player observes only their own successful placements and opponent stones discovered by attempting an already-occupied cell (a blocked move). We apply the *abrupt* rule. Rewards are zero during play; a successful placement that completes the acting player's connection immediately ends the game with $\pm 1$ (Hex admits no draws on a filled board). Observations are a length-9 vector with entries $\{-1, 0, 1\}$ denoting unknown, own stone, and discovered opponent stone.

**Abrupt Phantom Tic-Tac-Toe**: A two-player imperfect-information version of Tic-Tac-Toe on a $3 \times 3$ grid: players alternate attempting to place a mark but observe only their own successful placements and cells where a previous attempt was blocked (revealing an opponent mark). We also use the *abrupt* rule. Rewards are zero during play; a successful placement forming three-in-a-row yields $\pm 1$, and a full board with

no line results in a draw with 0 to both players. Observations are a length-9 vector with $\{-1, 0, 1\}$ encoding unknown, own mark, and discovered opponent mark.

**Liar's Dice**: We implement a single-hand version with 5 dice, each having 6 faces. This is a sequential bidding game in which each player privately rolls dice and alternately bids on the total count of a face value across both players until someone issues a challenge; the dice are then revealed to resolve the claim. Rewards are zero until a challenge occurs; the previous bidder wins if the claim is correct, and the challenger wins otherwise, yielding $\pm 1$. Observations contain 16 historical bidding states, each using three dimensions to record player ID, bid quantity, and bid face, plus six dimensions for the counts of each face in the player's private dice.

**Battleship**: A two-stage, two-player game on a $10 \times 10$ grid: players first place ships, then alternate attacks until one fleet is destroyed. Rewards are $+1$ for victory and $-1$ for defeat; termination occurs when a player's fleet is eliminated. Observations include 100 dimensions for the grid state, 10 for remaining ship statuses, and one for the current stage.

**Heads-Up No-Limit Texas Hold'em**: A two-player no-limit poker variant with blinds and four betting streets (preflop, flop, turn, river) and up to five community cards. Actions include folding, checking/calling, a 16 different fractional pot-sized raises, and all-in moves. At the beginning, the small blind/big blind is set to $1/2$ and stacks are initialized at 200 chips (100 big blinds) per player. Rewards are zero during play and correspond to normalized stack changes at the end of each hand; termination within a hand occurs upon a fold or at showdown (the overall match ends if a player goes bankrupt). Observations include two dimensions for the private and five for community cards, two for each player's stacks, two for each player's bets, one for pot size, one for the current street ID, and a sequence of historical actions. We use a history window of 64, each entry containing four dimensions (player ID, street ID, action taken, and betting amount).

## B.2 Number of Information States

For **Kuhn Poker** and **Leduc Poker**, the information-state counts are computed exactly using Open-Spiel's (Lanctot et al., 2019) built-in game-tree enumeration script (`count_states`), which traverses the full game tree and tallies distinct information states per player. For **Abrupt Dark Hex (3×3)** and **Abrupt Phantom Tic-Tac-Toe**, the counts are taken directly from Rudolph et al. (2025). For the three large-scale environments (**Liar's Dice**, **Battleship**, and **Heads-Up No-Limit Texas Hold'em**), we calculate ourself.

**Liar's Dice**: At any decision point, a player's information state is determined by (i) their private dice hand and (ii) the public bidding history. The hand is a multiset of 5 dice drawn from 6 faces, giving $\binom{5+6-1}{5} = 252$ distinct hands. Each bid is a (quantity, face) pair with quantity in $\{1, \ldots, 10\}$ and face in $\{1, \ldots, 6\}$, yielding 60 bids under a total lexicographic order; a challenge terminates the game. Because successive bids must be strictly increasing, a non-terminal public history is exactly a subset of the 60 bids, giving $\sum_{k=0}^{60} \binom{60}{k} = 2^{60}$ histories. Multiplying the two independent components yields

$$N_{\text{info}}^{\text{LD}} = 252 \cdot 2^{60} \approx 2.9 \times 10^{20}.$$

**Battleship**: At any non-terminal decision point in the attack phase, the player's information state contains (at minimum) two independent components: (i) their own ship placement, of which there are $A = 30{,}093{,}975{,}536$ for the standard fleet (ship lengths $\{5, 4, 3, 3, 2\}$, totaling $S = 17$ ship cells per player) on a $10 \times 10$ grid (, https://mathoverflow.net/users/27921/bill jestingrabbit), and (ii) an attack-result board against the opponent, which labels each of the 100 cells as *unattacked*, *miss*, or *hit*. To lower-bound the number of reachable attack-result boards, fix any single opponent placement $P$ covering 17 cells. For each subset $S \subseteq [100]$ of cells the player has attacked, the attack-result board is uniquely determined by $P$ (hits on $S \cap P$, misses on $S \setminus P$, unattacked on $[100] \setminus S$), and distinct $S$ yield distinct boards. The non-terminal constraint $|S \cap P| \leq 16$ excludes only $2^{83}$ of the $2^{100}$ subsets, giving at least $2^{83}$ reachable boards. Combining with the $A$ ship placements,

$$N_{\text{info}}^{\text{BS}} \geq A \cdot 2^{83} \approx 2.9 \times 10^{35}.$$

**Heads-Up No-Limit Texas Hold'em**: The full game of heads-up no-limit Texas Hold'em is vastly larger than any of the other environments considered here. In principle, players may wager any integer number of chips up to their entire stack, community cards can be combined with private hands in an astronomical number of ways, and stacks evolve continuously across hands, making an exact information-state count intractable. To obtain a concrete lower bound that reflects the complexity experienced by an agent operating under our experimental rules, we restrict to a fully discretized version of the game and enumerate its game tree exactly.

We enumerate information states under the standardized rule set used in our experiments: both players begin with 100 big blinds (BB), the small/big blind is 0.5/1 BB, and four betting streets are played (preflop, flop, turn, river). On the preflop, the small blind may limp, open to one of 13 discrete sizes (2.0–3.5 BB), or move all-in (15 first-action options in total). On any street, the acting player may check/call, choose among $F = 14$ fractional pot-size bets/raises $(20\%, 25\%, 33\%, 40\%, 50\%, 66\%, 75\%, 100\%, 125\%, 150\%, 175\%, 200\%, 300\%, 400\%$ of the current pot), move all-in, or fold; the total number of bets and raises per street is capped at $B = 16$. Since the discretized action space is a strict subset of the full no-limit action space, the resulting count is a lower bound on the true game complexity. A player's information state at any decision point is uniquely determined by (i) their two private hole cards, (ii) the community cards dealt so far, and (iii) the public betting history (which encodes both players' current chip stacks).

We count information states by exhaustively enumerating the game tree under these rules. To make the computation tractable, all chip amounts are rounded to the nearest $\frac{1}{12}$ BB before hashing; at this granularity, rounded states that differ correspond to genuinely distinguishable chip configurations, so no two distinct information sets are collapsed.

The enumeration proceeds in two phases. *Phase 1* walks the preflop betting tree with memoization, keying each decision node on the rounded (pot, effective stack, last-bet amount, cumulative bet count, acting-player role) tuple. Upon reaching the flop we record each unique rounded (pot, effective-stack) pair and the number of distinct preflop histories that lead to it, yielding 942 unique flop-entry configurations across 25,664 distinct preflop paths. *Phase 2* processes each flop-entry configuration independently: we build a directed acyclic graph (DAG) of postflop decision nodes spanning all three postflop streets, compute via topological sort the number of distinct root-to-node paths reaching every node, and weight the resulting path counts by the corresponding preflop history count before accumulating.

Let $D_s$ denote the total weighted decision-node count on street $s$ (summed over both players), and let $C_s$ denote the number of (hole-card, community-card) combinations observable by a single player upon entering that street:

$$C_{\text{pre}} = \binom{52}{2} = 1{,}326,$$
$$C_{\text{flop}} = \binom{52}{2}\binom{50}{3} = 25{,}989{,}600,$$
$$C_{\text{turn}} = \binom{52}{2}\binom{50}{3}\binom{47}{1} = 1{,}221{,}511{,}200,$$
$$C_{\text{river}} = \binom{52}{2}\binom{50}{3}\binom{47}{1}\binom{46}{1} = 56{,}189{,}515{,}200.$$

The total information-state count is $N_{\text{info}}^{\text{NLHE}} = \sum_s D_s \cdot C_s$. The per-street breakdown is:

| Street | Decision nodes $D_s$ | Card combinations $C_s$ | Information sets $D_s \cdot C_s$ |
|---|---|---|---|
| Preflop | 25,665 | 1,326 | $3.403 \times 10^7$ |
| Flop | 26,273,552 | 25,989,600 | $6.828 \times 10^{14}$ |
| Turn | 339,227,852 | 1,221,511,200 | $4.144 \times 10^{17}$ |
| River | 2,373,722,384 | 56,189,515,200 | $1.334 \times 10^{20}$ |
| **Total** | 2,739,249,453 | — | $1.338 \times 10^{20}$ |

$$N_{\text{info}}^{\text{NLHE}} \geq 1.338 \times 10^{20}.$$

## B.3 Algorithm Settings

We evaluate NASHPG with four established model-free baselines: Neural Fictitious Self-Play (NFSP) (Heinrich & Silver, 2016), Policy Space Response Oracles (PSRO) (Lanctot et al., 2017), Magnetic Mirror Descent (MMD) (Sokota et al., 2023) and Regularised Nash Dynamics (R-NaD) (Perolat et al., 2022). All methods except R-NaD employ PPO (Schulman et al., 2017) for best-response learning or policy updates, with hyperparameters listed in Table 4. Detailed implementation notes for each algorithm are provided below.

**NFSP.** Maintains a uniform mixture of policies comprising all previously trained agents. Each iteration, a new agent is initialized and uses PPO to train an approximate best-response agent against the mixture of prior policies. The newly trained agent is added to the population after each iteration.

**PSRO.** The setup follows NFSP but replaces uniform mixing with a meta-game constructed from pairwise interactions among historical agents. Each entry of the payoff matrix is estimated from rollouts with 128 parallel environments and 1,000 steps per environment. A Nash equilibrium of this meta-game is then approximated via fictitious play (Brown, 1951) for 10,000 iterations, and the resulting mixed strategy defines the new population distribution. Training then continues with a newly initialized agent. In addition, we train a distillation agent by behavior cloning on rollout actions generated by the opponent mixture, yielding a single policy trained from the mixture's induced behavior.

**MMD.** A single policy is trained in self-play using PPO. Our implementation follows the implementation in Rudolph et al. (2025). Based on their setup, the magnet policy is set to a uniform policy where we control the regularization strength by the entropy regularization, we set the magnet coefficient to 0.05 aligned with Sokota et al. (2023).

**R-NaD.** Our implementation of R-NaD is heavily inspired by Rudolph et al. (2025). Within each outer-loop iteration, a fixed anchor policy defines the regularization term and remains unchanged during the inner-loop updates. For each batch of self-play trajectories, we compute per-player V-trace targets under the regularized reward and optimize the sum of a critic regression loss and a Neural Replicator Dynamics (NeuRD) (Hennes et al., 2020) policy loss.

| Hyperparameter | Value |
|---|---|
| Optimizer | AdamW |
| Step size | $3.0 \times 10^{-4}$ |
| Num. PPO epochs | 4 |
| Num. minibatches | 4 |
| Discount ($\gamma$) | 1 |
| GAE parameter ($\lambda$) | 0.95 |
| Clipping parameter ($\epsilon$) | 0.2 |
| Entropy coeff. | 0.1 |
| Max grad norm | 0.5 |
| Number of envs. | 64 |
| Rollout length | 64 |

Table 4: Shared PPO hyperparameters

## B.4 Evaluation Metrics

In our work, we employ both *exploitability* and *head-to-head matches* to evaluate the policies of all baselines and NASHPG.

**Exploitability:** Consider a two-player zero-sum game with player's behavioural strategies $\pi_1$ and $\pi_2$. Let $\mu_i(\pi_i, \pi_{-i})$ denote the expected payoff for player $i$, where $-i$ denotes the opponent of player $i$. The best-response strategy for player $i$ is defined as

$$BR_i(\pi_{-i}) = \arg\max_{\pi_i'} \mu_i(\pi_i', \pi_{-i})$$

Given a suboptimal strategy $\pi_i$, the incentive for player $i$ to deviate is

$$\delta_i = \mu_i\big(BR_i(\pi_{-i}), \pi_{-i}\big) - \mu_i(\pi_i, \pi_{-i})$$

The exploitability is then defined as

$$\text{Exploitability} = \mathbb{E}_{i \sim \text{Uniform}}\big[\delta_i\big]$$

In our experiments, the procedure used to obtain $BR_i(\pi_{-i})$ depends on the environment. For Kuhn Poker and Leduc Poker, we compute exact exploitability using OpenSpiel's exploitability routine (Lanctot et al., 2019). We wrap our learned policy as an OpenSpiel-compatible behavioral policy that assigns probabilities to legal actions at each information state, and then compute the exact best-response gains for both players in the corresponding game tree. For Abrupt Dark Hex and Abrupt Phantom Tic-Tac-Toe, we use `exp-a-spiel` (Rudolph et al., 2025), a tabular solver, to compute exact exploitability. We enumerate all infosets, evaluate the learned policy at each infoset to obtain a full strategy table, and then derive the best-response policy and its value against the target policy via dynamic programming. For larger environments where exact game-tree computation is infeasible, we approximate the best-response value by training a newly initialized PPO exploiter against the fixed target policy at each evaluation checkpoint. In our large-environment experiments, the exploiter is trained for 10,000 updates. All other optimization and algorithmic hyperparameters follow the settings reported in Table 4.

**Head-to-Head Matches:** For large environments where exact exploitability is intractable, we complement approximate exploitability with direct pairwise evaluation. Specifically, the final policy of each baseline plays against the final policy of NASHPG, and we report the average payoff of NASHPG over 1,024 games. Each game is played with randomly assigned player positions to control for positional advantage. A positive average payoff indicates that NASHPG outperforms the baseline in direct play. All results are averaged over 5 random seeds and reported as mean $\pm$ standard deviation.

| Environment | Kuhn Poker | Leduc Poker | Dark Hex (3×3) | Phantom Tic-Tac-Toe | Liar's Dice | Battleship | Heads-Up Hold'em |
|---|---|---|---|---|---|---|---|
| Players | 2 | 2 | 2 | 2 | 2 | 2 | 2 |
| Obs. Dim. | 7 | 49 | 9 | 9 | 102 | 111 | 285 |
| Action spaces | 2 | 3 | 9 | 9 | 61 | 100 | 16 |
| Feature Extractor | Dense MLP | Dense MLP | Dense MLP | Dense MLP | Transformer | CNN | Transformer |
| Layers | 2 | 2 | 2 | 2 | 2 | 2 | 2 |
| Hidden Dim. | 16 | 64 | 64 | 64 | 64 | 128 | 512 |
| Attention Heads | – | – | – | – | 1 | – | 4 |
| History Length | – | – | – | – | 32 | – | 64 |
| Policy Head | Linear | Linear | Linear | Linear | Linear | CNN | Linear |
| Policy Layers | 1 | 1 | 1 | 1 | 2 | 2 | 1 |
| Critic Head | Linear | Linear | Linear | Linear | Linear | Linear | Linear |
| Critic Layers | 1 | 1 | 1 | 1 | 2 | 1 | 1 |
| Reward Design | $\Delta$ | $\Delta/20$ | $\pm 1$ | $\pm 1, 0$ | $\pm 1$ | $\pm 1$ | $\Delta/\text{Stack}$ |

Table 5: Summary of environments and model architectures

# C  Additional Experiment

## C.1  Anneal Regularization Strength as Nash Equilibria solver

Sokota et al. (2023) introduced *Magnetic Mirror Descent* (MMD) and suggested two potential strategies for applying MMD to compute Nash equilibria: (i) *moving magnet*, which interpolates between the magnet policy and the current policy, and (ii) *annealing the regularization strength*. While the moving magnet approach is conceptually appealing, it requires geometric blending of policies at every information set, which is computationally infeasible in large games. This leaves annealing as the more practical alternative.

Here, we examine the feasibility of annealing regularization strength in the presence of stochastic sampling noise, i.e., when gradients are estimated from sampled trajectories rather than exact feedback. We conduct

experiments on *Kuhn Poker*, a small imperfect-information game that exact computation of exploitability is feasible, making it a suitable testbed.

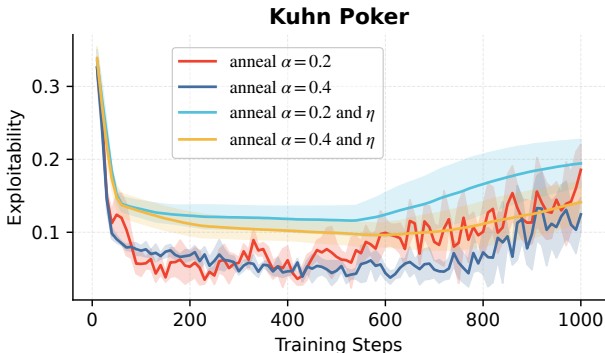

Figure 6: Exploitability under annealing different initial values for $\alpha$ (optionally annealing $\eta$). In all cases, exploitability diverges as $\alpha$ decreases.

We use the same PPO hyperparameters as all other experiments and investigate the effect of linearly decaying the magnet coefficient $\alpha$. Specifically, we test initial values $\alpha \in \{0.2, 0.4\}$, annealed linearly to 0.001. We additionally consider jointly decaying both $\alpha$ and the learning rate $\eta$, with $\eta$ decayed linearly to zero. Each setting is repeated over four runs, and we report the mean and standard deviation of exploitability. Results are shown in Figure 6.

The results reveal a consistent pattern: exploitability decreases steadily during the early stages of training, but as $\alpha$ becomes small, performance deteriorates and eventually diverges. Even when annealing $\eta$ jointly with $\alpha$, training proceeds more slowly and still exhibits divergence at later stages. These findings indicate that annealing regularization strength is inherently unstable in the stochastic setting. While careful hyperparameter tuning may mitigate these issues, the dependence of the dynamics on both $\alpha$ and $\eta$ (recall the MMD convergence constraint $\alpha \geq \mu\eta L^2$) makes annealing regularization an unattractive choice as a general-purpose solver for Nash equilibria.

