# OpenReview forum: "NashPG: A Policy Gradient Method with Iteratively Refined Regularization for Finding Nash Equilibria"
_TMLR — Under review for TMLR_

### Review · Reviewer_7kft · 2026-05-31

**Summary Of Contributions:**

The paper studies finding a Nash equilibrium in two-player zero-sum matrix games and proposes an iterative magnetic mirror descent algorithm, which regularizes the payoff with an additional Bregman divergence term evaluated using the algorithm iterate of the last iteration. The paper establishes that the last iterate of the algorithm converges asymptotically to a Nash equilibrium. They further generalize the algorithm to the practical, sample-based setting. The sample-based algorithm is evaluated empirically in a range of environments.

**Audience:**

Yes

**Audience Explanation:**

The work is of potential interest to researchers working on game theory and reinforcement learning.

**Broader Impact Concerns:**

No concerns.

**Claims And Evidence:**

Yes

**Claims Explanation:**

The paper's theoretical results are supported by complete analysis, and the technical assumptions are clearly stated. The empirical results are presented comprehensively across a broad set of games and provide convincing evidence.

**Requested Changes:**

The paper is overall organized and well-written, with the main contributions clearly presented. I believe the technical results are of interest to the game theory and RL community. A few questions and comments for the authors to consider:

1) The paper claims to target two-player zero-sum imperfect-information games. However, after drawing the connection to matrix games in Section 3.1, the authors focus the analysis on matrix games. Isn't it more natural to simply state the goal of the paper as solving matrix games?

2) The gap in the settings of Algorithm 3 and Algorithm 4 involves more than just replacing full-information updates with sample-based ones -- Algorithm 3 is in the matrix game setting, while Algorithm 4 aims to solve Markov games. This weakens the indirect theoretical support for Algorithm 4 from Theorem 3, as Theorem 3 does not account for the additional complexities introduced by state transitions in stochastic games.

3) The work conceptually relates very closely to another type of iterate regularization methods, which adds an identity/entropy term to the payoff function to introduce strong monotonicity/PL condition (for example, see Koshal et al. which solves a VI problem in the exact form considered in the present paper, and Zeng et al. which solves a more general stochastic game). The evaluation of such regularization terms does not involve the past algorithm iterate. However, the regularization weight needs to be dynamically reduced to zero for convergence to the original Nash equilibrium.

While the most natural (and conveniently analyzable) algorithm under entropy/identity regularization is perhaps a nested-loop one, these prior works show that the algorithm may also be streamlined into a single-loop form. Can the authors comment on whether the proposed algorithms in the present work may also be implemented in a single-loop manner, with inner loop $k$ only running for one iteration per outer-loop update, from both the theoretical and empirical perspectives?

4) What the asymptotic analysis does not reveal is how the complexity depends on $\alpha$. $\alpha$ only needs to be lower bounded for the analysis to go through, but it is not unexpected to see that as $\alpha$ becomes too large the convergence rate is negatively affected. It would be interesting to see if U-shaped curve in Figure 1 can be supported by mathematical analysis.

References

Koshal, J., Nedic, A. and Shanbhag, U.V., 2012. Regularized iterative stochastic approximation methods for stochastic variational inequality problems. IEEE Transactions on Automatic Control, 58(3), pp.594-609.

Zeng, S., Doan, T. and Romberg, J., 2022. Regularized gradient descent ascent for two-player zero-sum Markov games. Advances in Neural Information Processing Systems, 35, pp.34546-34558.

---

> ### Author Response · Authors · 2026-06-03
>
> We thank the reviewer for their thoughtful questions and constructive feedback. We address each point below.
>
> 1. The paper's goal is indeed to solve IIGs, not just matrix games. The matrix game setting is the natural starting point for theory, as it allows us to establish clean theoretical results (Sections 3--4.2) that serve as the foundation for the rest of the paper. Section 4.3 then develops NashPG via a series of approximations that adapt the iterative-refinement structure to the behavioral strategy space, and Section 5 evaluates NashPG on IIGs spanning several orders of magnitude in complexity.
>
> 2. We agree, and the approximation steps in Section 4.3 are best characterized as heuristic rather than formal derivation. Specifically, see the transition from the mixed-strategy objective (Eq. 11) to the behavioral-strategy analogue (Eq. 12). One might argue that, under realization-equivalence between mixed and behavioral strategies, these two objectives are equivalent; however, this is not the case. The regularization term in Eq. 12 is an expectation over observations sampled under the joint strategy profile $\pi$, making it depend on the opponent's strategy $\pi^{(-i)}$, whereas the Bregman divergence in Eq. 11 depends only on the player's own strategy.  This dependence on the opponent means the two objectives differ even when the payoff terms are realization-equivalent, so the theoretical chain is broken at this step. Consequently, as stated in the Remark at the end of Section 4.3, NashPG carries no formal convergence guarantee.  It is a theory-inspired algorithm whose effectiveness we validate empirically across a broad range of games.
>
> 3. This is an interesting connection. Decaying regularization was indeed the first approach we tried. However, the practical difficulty in the stochastic (sample-based) setting is that reducing $\alpha$ too much risks being overwhelmed by gradient estimation noise, causing training to destabilize and often diverge.  We include a small demonstration of this phenomenon in Appendix C.1: linear $\alpha$-annealing on Kuhn Poker consistently leads to divergence regardless of the initial $\alpha$ value, even when the learning rate $\eta$ is decayed jointly. By contrast, our multi-round approach keeps $\alpha$ fixed throughout training, so the stabilizing effect of regularization is preserved.
>
> Regarding implementing in a single-loop manner, we find this to be an interesting direction to explore.
>
> Empirically, we ran an experiment (not included in paper) on Rock-Paper-Scissors ($\alpha =
> 0.2$, both agents initialized at $[0.8,\,0.1,\,0.1]$) with $K=1$ inner
> iteration and observed that the joint policy simply cycles without converging.
> Adding momentum to the optimizer (e.g., Adam) caused divergence.  This
> suggests that a sufficient number of inner iterations is necessary for each outer-loop
> reference update to make meaningful progress toward equilibrium.
>
> Theoretically, our current analysis requires each inner VI subproblem to be solved to exact convergence.  An interesting open question is whether we can relax this: if the inner loop is only run for a finite $K$ steps and produces an approximate solution, does the outer-loop procedure still converge to a Nash equilibrium?  If so, one could then invoke the finite-time convergence results of MMD [1] to derive the number of inner iterations that are sufficient to reach the required approximation quality, turning the heuristic choice of $K$ into a principled one backed by theory.  We hope this direction can inspire future work.
>
> 4. We agree, and this aligns with our empirical results. A finite-time bound for IMMD would make this dependence explicit, as we expect $\alpha$ would naturally appear in such a bound, and could formally support the U-shaped curve in Figure 1.  We are unable to derive such a bound in the present work, and we hope it can be achieved in future work.
>
> [1] Sokota, Samuel, et al. "A unified approach to reinforcement learning, quantal response equilibria, and two-player zero-sum games." International Conference on Learning Representations 2023

---

### Review · Reviewer_7rXg · 2026-06-06

**Summary Of Contributions:**

The paper studies Nash equilibrium computation in two-player zero-sum imperfect-information games through iterative refinement of regularized objectives. Its main theoretical contribution is IMMD, a multi-round extension of MMD with monotonic Bregman-distance reduction and convergence to a Nash equilibrium under the stated assumptions. The practical contribution is NashPG, which translates the idea into a PPO-style policy-gradient algorithm by adding a KL term to the policy objective and periodically updating the reference policy.

Strengths: the theoretical framing is clean, the practical algorithm is simple, and the experiments cover both small exact-evaluation games and larger domains.

Weaknesses:

- the formal guarantee applies to IMMD, not NashPG
- the large-game evidence relies on approximate best responses and head-to-head play
- the empirical comparison does not fully establish performance against stronger domain-specific solvers

**Audience:**

Yes

**Audience Explanation:**

This paper should interest readers working on multi-agent RL, imperfect-information games, and scalable equilibrium computation.

**Broader Impact Concerns:**

I do not see major broader-impact concerns beyond those already noted by the authors.

**Claims And Evidence:**

Yes

**Claims Explanation:**

The main theoretical claim for IMMD is supported by a clear VI/Bregman-divergence argument, and the paper is careful to state that NashPG itself does not inherit the formal convergence guarantee. The empirical claims are mostly supported: NashPG achieves low or competitive exploitability in games where exact evaluation is possible, and the larger-domain results are supported by approximate best-response evaluation plus head-to-head matches.

**Requested Changes:**

- Clarify the wording around guarantees. The paper should consistently distinguish the convergence result for IMMD from the empirical NashPG algorithm, especially in the abstract/introduction where the "policy-gradient-based algorithm that provably converges" framing can be read too strongly.

- Strengthen the discussion of large-domain evaluation. Approximate BR and head-to-head results are useful, but the paper should state more clearly what they do and do not certify about exploitability.

- Provide more detail on the approximate best-response evaluation: exploiter training budget, sensitivity to exploiter strength, and whether stronger exploiters change the ranking.

---

> ### Author Response · Authors · 2026-06-23
>
> We thank the reviewer for the helpful suggestions. We address each point below.
>
>
> 1. The phrase ``policy-gradient-based algorithm that provably converges'' appears in Section 1 as a motivating research question, explicitly framed as a goal we set out to achieve, rather than a direct claim about NashPG. We answer this question in two parts: IMMD with formal convergence guarantees (Theorem 3), and NashPG as a scalable empirical algorithm. This distinction is maintained consistently throughout. Table 1 in related work explicitly marks NashPG as lacking convergence guarantees, and the concluding Remark of Section 4.3 reiterates that NashPG is a theory-inspired algorithm without formal guarantees. We therefore believe the current framing is carefully worded. That said, we are happy to revise any specific passages the reviewer found ambiguous or potentially overclaiming, and would welcome pointers to the exact wording of concern.
>
>
> 2. Thank you for the suggestion. We have further clarified in Section 5.1 what approximate exploitability measures. Because the RL-trained BR is not guaranteed to reach the true best-response value, the resulting approximate exploitability is a lower bound on true exploitability, where the tightness of the bound depends on the quality of the RL training.
>
>
> 3. Thank you for noting this. Each BR agent is trained for 10{,}000 update steps. All evaluated methods use the same BR training budget and setup for fairness. Regarding exploiter strength, the RL agent used as the approximate BR uses the same model size and architecture as the MARL agents, so both agents have the same representational capacity. We have added all of the above details to Section 5.1. Additionally, as evidence that approximate exploitability gives a useful signal of agent performance, Figure 4 and Table 3 show that the exploitability ranking broadly agrees the head-to-head performance ranking, which we discuss further in Section 5.3.

---

### Review · Reviewer_BSf6 · 2026-06-15

**Summary Of Contributions:**

The paper studies how to find Nash equilibria in two-player zero-sum (2p0s) imperfect-information games with a model-free, policy-gradient method.

Contributions.
1. IMMD: a multi-round scheme that solves a Bregman-regularized VI with operator $G_\rho(z)=F(z)+\alpha\nabla_z B_\psi(z;\rho)$ and resets the reference $\rho$ to the previous solution, i.e. $z_{t+1}=M(z_t)$. The authors prove monotone decrease of the Bregman divergence and asymptotic convergence to a Nash equilibrium.
2. NashPG: a practical algorithm that puts a per-observation KL term directly in the policy objective and can run on top of PPO.
3. Experiments on 7 games (Kuhn, Leduc, ADH3, APTTT, Liar's Dice, Battleship, Heads-Up NLHE), an ablation (RNaD-PPO), and new JAX implementations.

Strengths.
- The IMMD analysis looks correct and is clean.
- The paper is honest that NashPG has no formal guarantee.
- The method is simple and the empirical coverage is broad.

Weaknesses.
- A boundary-case gap in the convergence proof.
- The large-game exploitability metric is not valid; it becomes negative.
- Head-to-head payoff is used as equilibrium evidence, but it is not.
- Only 4 seeds, no significance test; some gains are within noise.
- The ablation conclusion is too strong.

**Audience:**

Yes

**Audience Explanation:**

The topic is of interest to TMLR readers in multi-agent RL, game theory, and equilibrium computation: how to scale regularization-based last-iterate methods with standard policy gradients. Several parts are useful even if the contested claims are removed: the IMMD operator view and its analysis, a simple method, the RNaD-PPO ablation, and new JAX implementations of several large games.

**Broader Impact Concerns:**

No additional concern.

**Claims And Evidence:**

No

**Claims Explanation:**

The paper is honest about what is proven and what is heuristic, and the small-game results with exact exploitability support the claim there. But several main claims are not yet well supported. I have four main concerns below:

1. The convergence proof has a boundary gap. Lemma 1 (monotone decrease) is fine. But the convergence part takes a subsequence limit $z_\infty$ and then uses $M$ and $\nabla\psi$ at $z_\infty$. For the negative-entropy $\psi$ that the method uses, $\nabla\psi=\log$ blows up on the simplex boundary, and $M$ is only defined on $\mathrm{int}\,\mathrm{dom}\,\psi$ (Def. 2). Finite 2p0s games often have equilibria on the boundary (e.g. Kuhn), so this is the common case, not a corner case.

2. The large-game exploitability is not questionable. By the definition in App. B.4, exploitability is non-negative. But the curves go to about $-0.1$ to $-0.2$. This means the PPO best-responder (10k updates) is too weak and does not even reach the policy's own value. So these plots do not bound the true exploitability in the large games, which are exactly where scalability is claimed.

3. Head-to-head is not equilibrium evidence, and some gaps are within noise. In particular, head-to-head payoff measures relative strength, not distance to Nash; a more exploitable policy can still win. Also, with 4 seeds, the gap to MMD is within noise: Liar's Dice $+0.010\pm0.040$ and Battleship $+0.006\pm0.006$. So "higher average payoff" in the abstract is too strong.

4. The ablation claim in Sec. 5.4 is too strong. The claim that the bottleneck is "PPO vs NeuRD" goes against the paper's own Fig. 5a,b: in Kuhn and Leduc, NeuRD is better than RNaD-PPO. So the effect depends on the regime.

If they are addressed, I would change this answer to Yes.

**Requested Changes:**

1. Fix the large-game exploitability metric. Negative values should not be possible for exploitability. Please (a) report the approximate-BR value and the policy's own value separately, (b) show exploiter training curves to confirm it converged (10k updates seems too few for NLHE/Battleship), (c) use the strongest exploiter, and (d) call it a lower bound, not "exploitability".

2. Fix or restrict Theorem 3. Handle the boundary case where $z_\infty$ is on the simplex boundary (where $\nabla\psi$ and $M$ are not defined), or restrict the theorem to interior equilibria and show the iterates stay in a compact interior set. Lemma 1 is not affected.

3. Calibrate the head-to-head and the statistics. Present Table 3 as relative strength, not as equilibrium evidence. Add significance tests or confidence intervals over seeds, and mark the within-noise cells (vs MMD on Liar's Dice and Battleship). Soften the abstract wording.

4. Relate IMMD to the classical Bregman proximal-point method for monotone VIs. This is about correct attribution and position of this work, not novelty.

5. Justify Eq. 11 (first-order equivalence) and Eq. 12 (behavioral KL vs mixed-strategy KL); note that the KL in Eq. 12 depends on the opponent and on the visitation under $\pi$.

6. Add or scope the model-free CFR baselines (DREAM, ESCHER), which are discussed but not compared.

---

> ### Author Response · Authors · 2026-06-23
>
> We thank the reviewer for the thorough reading of our paper, including the appendix proofs. We address each concern and change request below.
>
> > Concern 1 / Requested Change 2: Convergence proof has a boundary gap
>
> The reviewer is correct to identify this gap. In the convergence proof, the subsequence
> $\{z_t\}$ lies in sublevel set $S = \{z : B_\psi(z^{\*};z) \leq B_\psi(z^{\*};z_0)\}$,
> but the limit $z_\infty$ is not guaranteed to lie in $\operatorname{int}\operatorname{dom}\psi \cap \mathcal{Z}$.
> We have fixed this by adding the assumption $z^{\*} \in \operatorname{int}\operatorname{dom}\psi \cap \mathcal{Z}$ to Theorem 3 and Lemma 2.
> We note that IMMD does not prescribe a specific $\psi$; choosing $\psi = \ell_2$
> (with full domain $\mathbb{R}^n$) satisfies $z^{\*} \in \operatorname{int}\operatorname{dom}\psi \cap \mathcal{Z}$
> trivially for any game.
> Using negative entropy (KL regularization) in NashPG is a deliberate design choice,
> as KL is the canonical regularizer between distributions in RL
> [1,2];
> however, as the reviewer notes, many games have Nash equilibria on the simplex boundary,
> violating $z^{\*} \in \operatorname{int}\operatorname{dom}\psi \cap \mathcal{Z}$.
> We have added this to the remark at the end of Section 4.3 as an explicit reason why
> NashPG does not carry a formal convergence guarantee.
>
> [1] Schulman, John, et al. "Trust region policy optimization." International Conference on Machine Learning. 2015.
>
> [2] Jacob, Athul Paul, et al. "Modeling strong and human-like gameplay with KL-regularized search." International Conference on Machine Learning 2022.
>
> > Concern 2 / Requested Change 1: Large-game exploitability is not valid
>
> Thank you for pointing this out.
> We agree that _exact_ exploitability is non-negative by definition; however, what
> we report is _approximate_ exploitability computed via an RL-based best-responder,
> which can yield negative values when the BR agent is weaker than the evaluated policy.
> This is a known property of the metric where negative values arise when the BR fails to
> surpass the policy's own value.
> As a concrete illustration, in an early version of our Battleship environment with
> sparse $\pm 1$ terminal rewards, the approximate exploitability for a strong
> NashPG agent quickly dropped to $-1$ after only a few outer iterations,
> because a random BR agent facing a strong opponent has near-zero chance of winning
> and thus never improves past the initial value.
> We addressed this by switching to fine-grained partial rewards (rewarding each hit
> on a target cell), which gives the BR agent a learnable signal and produces the
> smoother curves shown in Figure 4.
>
> To our knowledge, approximate exploitability using an RL-based approximate BR is the standard approach for
> measuring equilibrium distance in large games where exact computation is intractable,
> and is adopted in prior work [3,4,5,6].
> We agree with the reviewer that these values constitute a lower bound on true
> exploitability, and have updated Figure 4 and Section 5.1 to
> explicitly state it as a lower bound,
> following prior work's terminology of ``approximate exploitability'',
> and add more detail on the BR setup (model architecture, training budget). All changes in the revised manuscript are highlighted in blue colour.
>
> We also note that in our definition of (approximate) exploitability (App. B.4) the per-player policy values $\mu_i(\pi_i,\pi_{-i})$ cancel out when averaging $\delta_1$ and $\delta_2$ in a two-player zero-sum game, so the reported metric depends only on the (approximate) BR values, not on the policy values themselves.
> We therefore believe approximate exploitability as a single scalar is the most interpretable summary of equilibrium distance and is consistent with prior work [3,4,5,6]; that said, we are happy to additionally report the per-player values in the appendix if the reviewer believes it would be informative.
>
> Regarding the 10k update budget and exploiter training curves, this is mainly constrained by computation cost.
> With 3 large games, 5 methods, 4 seeds, and 25 data points each, we conduct
> 1,500 BR training runs in total; for NLHE and Battleship, a single 10k-step PPO
> run takes approximately 6 hours.
> We believe a larger budget would yield a higher BR value, but scaling further
> is prohibitively expensive, and the large number of runs makes it hard to
> store or display individual training curves.
> Nevertheless, the 10k-step BR still provides a meaningful signal, Figure 4 shows
> that approximate exploitability is stable across consecutive outer iterations and
> that different methods produce clearly separated curves.
> If the BR were not learning at all, we would expect high variance between adjacent
> data points rather than the smooth, consistent trends observed.
> We therefore believe the metric, while a lower bound, provides a meaningful and
> fair comparison across methods.

---

> ### Author Response · Authors · 2026-06-23
>
> Regarding using the strongest exploiter, the strongest exploiter is policy-dependent, each checkpoint would require its own tuned best-responder.
> Using per-method or per-checkpoint tuning would compromise the fairness of the comparison, since differences in approximate exploitability would then reflect differences in the exploiter setup rather than in the evaluated policies.
> For this reason, following prior work [3,4,5,6], we adopt a standardized BR procedure where the same architecture (matching the policy agents), the same training budget, and the same random initialization applied across all methods and checkpoints.
>
>
> [3] McAleer, Stephen, et al. ``XDO: A Double Oracle Algorithm for Extensive-Form Games.'' Advances in Neural Information Processing Systems 2021.
>
> [4] McAleer, Stephen, et al. ``Anytime PSRO for Two-Player Zero-Sum Games.'' Reinforcement Learning in Games workshop 2022.
>
> [5] Steinberger, Eric, et al. ``DREAM: Deep Regret Minimization with Advantage Baselines and Model-Free Learning.'' arXiv 2020.
>
> [6] Timbers, Finbarr, et al. ``Approximate Exploitability: Learning a Best Response in Large Games.'' Proceedings of the Thirty-First International Joint Conference on Artificial Intelligence, 2022.
>
> > Concern 3 / Requested Change 3: Head-to-head is not equilibrium evidence, and some gaps are within noise
>
> Thank you for this observation.
> We agree that head-to-head payoff measures relative strength, not distance to Nash,
> and we did not intend to present it as equilibrium evidence, a more exploitable policy
> can indeed still win head-to-head.
> The reason we include head-to-head results is that this is common practice in prior work
> when comparing MARL algorithms, and it addresses a natural reader question: how does
> NashPG perform when playing directly against other trained agents?
> We have updated Section 5.3 to frame Table 3 more explicitly as a relative-strength
> comparison rather than an equilibrium distance measure.
>
> We also agree that for Liar's Dice and Battleship against MMD, the gap falls within
> one standard deviation of zero, indicating the result is not statistically reliable.
> We ran only 4 seeds due to computation constraints; with this sample size, formal
> significance tests would have very low power and are therefore not informative.
> That said, to provide stronger statistical support for our comparisons, we have additionally run one more seed for each method, bringing the total to 5 seeds in the revised paper.
> We believe reporting mean $\pm$ std is the most appropriate summary available, and have updated
> Table 3 to mark these within-noise cells explicitly.
> We have also updated the abstract, replacing "higher average payoff" with
> "comparable or higher average payoff" to reflect this.
>
> > Concern 4: Ablation claim in Sec. 5.4 is too strong
>
> Thank you for pointing this out. We agree the concluding claim in Section 5.4 was
> stated too broadly.
> The paper already notes (Section 5.4, paragraph 2) that _``in the smallest
> games, including Kuhn Poker and Leduc Poker, replacing NeuRD with PPO can hurt
> performance''_, so the regime dependence is documented in our results.
> The intuition is that in small games, optimization is already sufficiently stable
> that the theoretical structure of the update matters more than practical heuristics,
> giving NeuRD an advantage.
> In larger and more challenging games, however, PPO's practical optimization advantages
> (clipped surrogate objective, value function baseline, etc.) become decisive.
> We have softened the concluding claim in Section 5.4 to make clear that the bottleneck
> finding applies specifically to the larger-game regime.
>
> > Requested Change 4: Relate IMMD to the classical Bregman proximal-point method for monotone VIs
>
> Thank you for pointing this out, we were not aware of this connection and appreciate the observation. We have added a paragraph in Section 4.2 making this connection explicit with appropriate citations.
>
> > Requested Change 5: Justify Eq. 11 (first-order equivalence) and Eq. 12 (behavioral KL vs. mixed-strategy KL)
>
> For Eq. 11, we have added Proposition 1 in the appendix with a formal proof. We also acknowledge that the original text said "gradient ascent" where it should more precisely say "natural gradient ascent" on the simplex; we have corrected this in the updated paper.
> For Eq. 12, we agree the behavioral KL and the mixed-strategy KL are not the same objective, and the paper already states this explicitly as a heuristic element. The reviewer is also correct that the observation visitation depends on both $\pi^{(2)}$ and $\pi^{(1)}$ itself; we have updated Section 4.3 to make both sources of discrepancy more explicit. Together, these gaps are exactly why NashPG does not carry a formal convergence guarantee, as stated in the Remark at the end of Section 4.3.

---

> ### Author Response · Authors · 2026-06-23
>
> > Requested Change 6: Add or scope the model-free CFR baselines (DREAM, ESCHER)
>
> We scope our experimental comparison rather than add new baselines.
> Recent benchmarking [7] shows that model-free CFR methods including DREAM and ESCHER underperform MMD and R-NaD on standard IIG benchmarks, the same methods we already compare against; ESCHER [8] also shows it improves over DREAM and NFSP.
> Based on these prior results, we do not expect these methods to outperform NashPG; however, we note that [7] focuses on smaller benchmark games, and a direct comparison in large games would be a valuable direction for future work.
> We have added a sentence to the related work section making this scoping explicit.
>
> [7] Rudolph, Max, et al. ``Reevaluating Policy Gradient Methods for Imperfect-Information Games.'' International Conference on Learning Representations 2026.
>
> [8] McAleer, Stephen, et al. ``ESCHER: Eschewing Importance Sampling in Games by Computing a History Value Function to Estimate Regret.'' International Conference on Learning Representations 2023.

---

### Author Response · Authors · 2026-06-23

We thank all reviewers for their careful and thorough reading of our paper.
The feedback has been insightful and helped us identify aspects of our work that required clarification or correction; we believe the paper is substantially stronger as a result. We have revised the manuscript accordingly, with all changes highlighted in blue text.
The most notable updates are:
1. **Theorem 3 and Lemma 2 (convergence):** We have added the assumption that the Nash equilibrium $z^{\*}$ lies in the interior of the domain of the Bregman generating function $\psi$, and revised the proof accordingly.
2. **Additional seeds:** We have run one additional seed for each method and game, bringing the total to 5 seeds, to provide stronger statistical support for our comparisons.
3. **Figure 3 correction:** The original manuscript contained an error in which the curves for NFSP and PSRO were shifted by one data point; this has been corrected in the revised manuscript.
4. **Approximate exploitability (Figure 4):** We have added further detail in Section 5.1 on how approximate exploitability is computed, and relabelled Figure 4 to use the term "approximate exploitability" rather than "exploitability" to more accurately reflect that these values constitute a lower bound on true exploitability.

We hope these revisions adequately address the reviewers' concerns.
Please do not hesitate to raise any further questions or points of discussion.